# Anti-Inflammatory and Cancer-Preventive Potential of Chamomile (*Matricaria chamomilla* L.): A Comprehensive In Silico and In Vitro Study

**DOI:** 10.3390/biomedicines12071484

**Published:** 2024-07-05

**Authors:** Assia I. Drif, Rümeysa Yücer, Roxana Damiescu, Nadeen T. Ali, Tobias H. Abu Hagar, Bharati Avula, Ikhlas A. Khan, Thomas Efferth

**Affiliations:** 1Department of Pharmaceutical Biology, Institute of Pharmaceutical and Biomedical Sciences, Johannes Gutenberg University, Staudinger Weg 5, 55128 Mainz, Germany; adrif@uni-mainz.de (A.I.D.); ryuecer@students.uni-mainz.de (R.Y.); r.damiescu@uni-mainz.de (R.D.); neltayeb@unimainz.de (N.T.A.); 2National Center for Natural Products Research (NCNPR), School of Pharmacy, University of Mississippi, Oxford, MS 38677, USA; bavula@olemiss.edu (B.A.); ikhan@olemiss.edu (I.A.K.)

**Keywords:** anti-inflammatory, carcinogenesis, cytokines, flavonoids, natural products, prevention, proteomics, Kaplan–Meier survival analysis, microscale thermophoresis

## Abstract

Background and aim: Chamomile tea, renowned for its exquisite taste, has been appreciated for centuries not only for its flavor but also for its myriad health benefits. In this study, we investigated the preventive potential of chamomile (*Matricaria chamomilla* L.) towards cancer by focusing on its anti-inflammatory activity. Methods and results: A virtual drug screening of 212 phytochemicals from chamomile revealed β-amyrin, β-eudesmol, β-sitosterol, apigenin, daucosterol, and myricetin as potent NF-κB inhibitors. The in silico results were verified through microscale thermophoresis, reporter cell line experiments, and flow cytometric determination of reactive oxygen species and mitochondrial membrane potential. An oncobiogram generated through comparison of 91 anticancer agents with known modes of action using the NCI tumor cell line panel revealed significant relationships of cytotoxic chamomile compounds, lupeol, and quercetin to microtubule inhibitors. This hypothesis was verified by confocal microscopy using α-tubulin-GFP-transfected U2OS cells and molecular docking of lupeol and quercetin to tubulins. Both compounds induced G2/M cell cycle arrest and necrosis rather than apoptosis. Interestingly, lupeol and quercetin were not involved in major mechanisms of resistance to established anticancer drugs (ABC transporters, *TP53*, or *EGFR*). Performing hierarchical cluster analyses of proteomic expression data of the NCI cell line panel identified two sets of 40 proteins determining sensitivity and resistance to lupeol and quercetin, further pointing to the multi-specific nature of chamomile compounds. Furthermore, lupeol, quercetin, and β-amyrin inhibited the mRNA expression of the proinflammatory cytokines *IL-1β* and *IL6* in NF-κB reporter cells (HEK-Blue Null1). Moreover, Kaplan–Meier-based survival analyses with NF-κB as the target protein of these compounds were performed by mining the TCGA-based KM-Plotter repository with 7489 cancer patients. Renal clear cell carcinomas (grade 3, low mutational rate, low neoantigen load) were significantly associated with shorter survival of patients, indicating that these subgroups of tumors might benefit from NF-κB inhibition by chamomile compounds. Conclusion: This study revealed the potential of chamomile, positioning it as a promising preventive agent against inflammation and cancer. Further research and clinical studies are recommended.

## 1. Introduction

It is well-known that inflammation can promote tumorigenesis [1]. Numerous clinical investigations have unraveled that precancerous disorders resulting from inflammation can be prevented from malignant progression by inhibiting inflammation [2]. Therefore, the quest for novel, effective options to prevent inflammation-related carcinogenesis represents a major focus in oncological research.

In recent years, natural products have gained considerable interest because of their diverse chemical composition, good tolerability, and beneficial pharmacological properties. It has been repeatedly reported that a majority of the current anticancer drugs established in the clinic are of natural origin or derived from natural products [3,4,5]. Furthermore, advancements in analytical tools and bioinformatics (e.g., “-omics” technologies) have made it easier to harness the potential of natural compounds [3,6,7,8]. One of the most renowned examples is paclitaxel, extracted from the bark of the Pacific yew, *Taxus brevifolia* [9]. Another notable example is artemisinin, obtained from *Artemisia annua,* which exerts not only antimalarial but also anti-inflammatory and anticancer activities [10]. Similarly, chamomile represents another example of a valuable medicinal plant. It is not just a well-known and widely consumed beverage, as it also possesses pharmacological properties due to its rich content of phenols and flavonoids, which confer antioxidant, antiproliferative, anti-inflammatory, and potential anticancer effects [11,12,13]. Given the established connection between inflammation and the development of cancer, we focused on the anti-inflammatory properties of *Matricaria chamomilla.*

Previously, we analyzed the inhibitory effects of chamomile on cyclooxygenase-2 (COX2) [12]. Because COX2 expression is regulated by nuclear factor kappa B cells (NF-κB), we then shifted our focus to this transcription factor. COX2 and NF-κB are both well-known players in the inflammatory process. COX2 is an enzyme responsible for producing prostaglandins during inflammation [14]. Nuclear factor-kB is crucial for various cellular processes including inflammation, immunity, cell growth, differentiation, and apoptosis [15]. The NF-κB pathway is generally a proinflammatory signaling pathway by promoting the expression of proinflammatory genes, such as cytokines, chemokines, and adhesion molecules [16]. It is activated through the conversion of the IKB kinase complex (IKK) into a catalytic active form, leading to the degradation of the IKB–NF-κB complex, which releases NF-κB into the nucleus [17]. In chronic inflammation and cancer, NF-κB is persistently dysregulated and active, and it has been implicated in the promotion of cell growth, angiogenesis, and cell proliferation [18]. This underscores the importance of the discovery and development of drugs targeting NF-κB and its pathways.

Thus, our aim in this study was to investigate the inhibitory activity of the secondary metabolites of chamomile (*Matricaria chamomilla* L.) against NF-κB. Through in silico screening, we have determined 6 out of 212 chamomile compounds based on their binding energy (kcal/mol) by using bioinformatical compound screening with PyRx and molecular docking with AutoDock4.2.6. Given our prior research on apigenin [19,20], we now directed our focus towards β-amyrin. The inhibitory activity against NF-κB was evaluated using microscale thermophoresis, an NF-κB reporter cell assay, as well as flow cytometric measurements of reactive oxygen species and mitochondrial membrane potential.

In the second part of our analyses, we selected the cytotoxic compounds lupeol and quercetin from *M. chamomilla*. Using oncobiogram analyses with the NCI panel of tumor cell lines and subsequent verification through confocal microscopy with α-tubulin-GF- transfected U2OS cells, we found that both compounds inhibited microtubules and induced G2/M cell cycle arrest. To perform hierarchical cluster analyses of the proteomes of 60 NCI cell lines, we generated a bioactivity oncobiogram. As lupeol and quercetin were not involved in classical drug resistance mechanisms (e.g., ABC transporters, *TP53*, *EGFR*), hierarchical cluster analyses using proteomic expression data of 3171 proteins in the NCI tumor cell line panel identified candidate proteins predicting sensitivity or resistance to these two compounds.

Moreover, Kaplan–Meier-based survival analyses with NF-κB as the target protein of these compounds were performed by mining the TCGA-based KM-Plotter repository with 7489 cancer patients to envision which tumor types and subtypes might benefit from chamomile treatment.

## 2. Materials and Methods

### 2.1. Phytochemical Analysis, Virtual Drug Screening, and Molecular Docking

Phytochemical analysis through liquid chromatography–diode array detector–quadrupole time-of-flight mass spectrometry (LC-DAD-QToF) was previously reported by us [12]. Virtual drug screening of more than 1000 chamomile compounds to COX2 has been also reported by us [12]. This repository of compounds was used for virtual screening and molecular docking to NF-κB in the present investigation.

The crystal structure of NF-κB was taken from the Protein Data Bank (PDB: 1NFI) [21,22], and only the homodimer structure of NF-κB p65-RelA was prepared for virtual drug screening using the software ChimeraXV5 (University of California in San Francisco, San Francisco, CA, USA) (accessed on 30 March 2023) [23]. The water molecules were deleted, polar hydrogen atoms were added and merged, the missing atoms and bonds were repaired, and Kollman charges were added. More than 1000 chamomile compounds were then screened for binding to NF-κB p65-RelA using PyRx version 0.8 [24,25]. Afterwards, 212 out of the 1000 ligands were selected based on their lowest binding energy (LBE, kcal/mol) and subjected to molecular docking against NF-κB using AutoDock 4.2.6 [26,27,28]. The DNA active site situated in the Rel homology domain (RHD) of NF-κB was defined for the grid box [17,21,29]. Its dimensions were 38 × 58 × 82 Å spacing, 0.622 Å at the grid-center, x = −4.515 Å, y = 72.657 Å, and z = 100.06 Å. The Lamarckian Genetic Algorithm (LGA) was applied to seek the lowest binding energies (LBE, kcal/mol) and predicted inhibition constants (pKi, µM) with docking parameters set to 250 runs and 2,500,000 energy evaluations for each cycle. The data from the histogram were organized in an Excel table (Microsoft Excel 2021 (Version 2306, Build 16.0.16529.20164)). The mean values ± SD were calculated from each of the three independent dockings. BIOVIA Discovery Studio Visualizer 2021 was used to generate the 3D visualization of interactions between the ligand and the amino acid residues of the crystal structure 1NFI [30].

### 2.2. Microscale Thermophoresis

The microscale thermophoresis (MST) technique was performed to validate the interaction between NF-κB (12054-H09E, Sino Biological Europe GmbH, Eschborn, Germany) and β-amyrin. The recombinant NF-κB protein was labeled using the Monolith Protein Labeling Kit RED-NHS 2nd Generation (MO-L011, NanoTemper Technologies GmbH, Munich, Germany) in accordance with the protocol provided by the manufacturer. Various concentrations (starting from 300 µM to 30 nM) of β-amyrin were incubated with labeled NF-κB (at a concentration of 200 nM) (1:1) for 30 min at room temperature in the dark. The analysis was performed using standard capillaries in the Monolith NT.115 system (NanoTemper Technologies GmbH, Munich, Germany). The MST experiment was conducted with an LED power of 40% and an MST power of 10% for the labeled NF-κB. For the data analysis, the NanoTemper Analysis Software was utilized.

### 2.3. Cell Culture

The HEK-Blue Null1 cells are a subtype of human embryonic kidney cells (HEK 293) that express secreted embryonic alkaline phosphatase (SEAP) under the control of an NF-κB promoter. The cells were obtained from Invivogen (Toulouse, France) (https://www.invivogen.com/hek-blue-null1v, accessed on 5 December 2019). The cells were maintained in DMEM medium supplied with 2 mM of L-glutamine, 10% fetal bovine serum (FBS), 1% penicillin–streptomycin (Invitrogen, Darmstadt, Germany), and 1 mL of normocin (100 µg/mL). After the second passage, Zeocin (100 µg/mL) was then added to the media (Invivogen, Toulouse, France). The cells were cultured at 37 °C in a humidified environment with 5% CO_2_ [31].

The drug-sensitive CCRF-CEM and multidrug-resistant P-glycoprotein-overexpressing CEM-ADR5000 leukemia cells were cultured in RPMI medium mixed with 1% penicillin/streptomycin and 10% FBS. Doxorubicin was added to CEM/ADR5000 every two weeks. The HCT116 p53^+/+^ human wild-type colon cancer cells and their knockout p53^−/−^, alongside the wild-type human glioblastoma U87.MG cells and their transfected cells with ΔEGFR (U87.MGΔEGFR), were all nurtured in DMEM medium mixed with 1% penicillin/streptomycin and 10% FBS. U87.MGΔEGFR and HCT116 p53^−/−^ were treated every two weeks with geneticin (400 µg/mL).

### 2.4. NF-κB Reporter Assay

The assay was performed according to the instructions of Invivogen (Toulouse, France). The HEK-Blue Null1 (HKBN1) cells were seeded in a 96-well plate at 100 µL/well and 50,000 cells/well overnight. Then, cells were treated for 24 h with 0.1 µM, 1 µM, and 10 µM of β-amyrin, β-sitosterol, β-eudesmol, daucosterol, myricetin, and apigenin. Triptolide was used as the positive control at concentrations of 0.1 µM and 1 µM. DMSO served as a negative control. TNF-α (100 ng/mL) was added for 24 h to induce the activity of NF-κB. The incubation was at 37 °C in 5% CO_2_. QUANTI-Blue™ (QB) solution was used for the detection and quantification of NF-κB (Invivogen, Toulouse, France), and 180 µL of QB was mixed with 20 µL of the supernatant and then incubated for 15 min at 37 °C in 5% CO_2_ [32]. The measurement was at 620–655 nm using a microplate reader (Tecan, Crailsheim, Germany). The experiment was independently repeated three times.

### 2.5. Mitochondrial Membrane Potential Assay

The JC-1 mitochondrial membrane potential assay kit was purchased from Cayman Chemical (Distributor Biomol GmbH, Hamburg, Germany) [33] and used following their instructions [34]. Aliquots of 125,000 HBN1 cells/well and 2 mL/well were seeded in a 6-well plate and left overnight. β-Amyrin (10 µM) and vinblastine as the positive control were added to the cells along with DMSO as the negative control for 24 h. The next day, 100 ng/mL of TNF-α were added for 3 h to induce the activation of NF-κB. The cells were then stained with prediluted JC-1 (1 µL of JC-1 in 9 µL of culture medium) and incubated at 37 °C for 15 min in the dark. Next, the cells were washed with cell-based assay buffer and centrifuged twice at 400× *g* for 5 min. Finally, the samples were directly measured with a flow cytometer (Novocyte Quanteon, Agilent Technologies, Frankfurt, Germany). Samples of 20,000 cells were analyzed and separated depending on the fluorescence intensity. The JC-1 dye was activated using a 488 nm argon laser. The JC-1 aggregates and monomers both emit green fluorescence (measured at 527 nm), which is detected in the FL1 channel (set at 530 nm). However, JC-1 aggregates representing the healthy cells also emit red fluorescence (measured at 595 nm), detected in the FL2 channel (set at 590 nm) [35]. All experiments were performed three times. The FSC files were analyzed using FlowJo_V10 Software (FLOWJO.LLC 1997–2018).

### 2.6. ROS Detection

The HEK-Blue null1 cells were seeded in a 6-well plate and incubated for 24 h, allowing attachment. The treatment was for 24 h with varying concentrations of β-amyrin (0.1 µM, 1 µM, and 10 µM). The 3rd day, 100 ng/mL of TNFα was added and left for 24 h. The 4th day, the cells were harvested, washed, and suspended with 1 mL of PBS. 2’7-Dichlorodihydrofluorescein diacetate (H_2_DCFH-DA, 10 μM; Sigma-Aldrich, Taufkirchen, Germany) was added and incubated for 30 min at 37 °C. Cells were treated with H_2_O_2_ (10 µL of the stock concentration; Sigma-Aldrich) for 15 min to activate the ROS generation. DMSO plus H_2_O_2_ plus TNF-α, cells plus H_2_O_2_, as well as cells without any treatment were used as controls. Lastly, the samples were directly assessed using a Novocyte Quanteon flow cytometer (Agilent Technologies). The analysis was performed with FlowJo Software (FLOWJO.LLC, 1997–2018). The procedure was independently conducted in triplicate, as described by us [36,37].

### 2.7. Growth Inhibition Assays

The β-amyrin cytotoxicity was measured through the resazurin reduction assay at 7 different concentrations. Human CCRF-CEM leukemia cells were seeded at a density of 1 × 10^4^ cells/well and 100 µL/well of RPMI 1640 medium in 96-well plates. The cytotoxicity of lupeol and quercetin (Sigma–Aldrich) was tested at 10 µM, 25 µM, 50 µM, and 100 µM on six different cell lines: CCRF-CEM, CEM/ADR5000, U87.MG, U87.MG/ΔEGFR, HCT116 p53^+/+^, HCT116 p53^−/−^, and U2OS.

After 72 h of treatment with β-amyrin, lupeol, and quercetin, 20 µL of 0.01% resazurin solution from Promega (Mannheim, Germany) was added to each well and incubated for over 4 h. The fluorescence signal was subsequently detected using an Infinite M2000 Pro-plate reader (Tecan) with an excitation wavelength of 544 nm and an emission wavelength of 590 nm. The experiment was performed independently in triplicate, and the concentrations were tested in sextuplicate. The results were interpreted as a percentage of cell viability and graphed as a dose–response curve. The IC_50_ value was calculated using Microsoft Excel 2021 (Version 2306, Build 16.0.16529.20164).

A range of human tumor cell lines with diverse origins, including leukemia, melanoma, brain tumors, and carcinoma of the lung, colon, kidney, ovary, breast, or prostate were employed by The National Cancer Institute’s Developmental Therapeutics Program (Bethesda, MA, USA) [38,39] to conduct drug screening. The Log_10_IC_50_ values, obtained through a sulforhodamine 123 assay, along with transcriptomic and proteomic expression data, were deposited on the NCI website [40]. Statistical correlation analyses were performed using Pearson’s correlation test (WinStat, Kalmia Inc., Cambridge, MA, USA; accessed on 19 November 2023).

### 2.8. Immunofluorescence Microscopy of α-Tubulin

U2OS osteosarcoma cells were cultured in μ-slide 8-well plates (ibidi, Gräfelfing, Germany) at a density of 30,000 cells per well and allowed to adhere for 24 h. Subsequently, the cells were treated with concentrations of 0.1, 1, and 10 μM of β-amyrin, quercetin, and lupeol. As positive controls, paclitaxel (1 μM) and vincristine (1 μM) were used (obtained from the University Hospital Pharmacy, Mainz, Germany), while DMSO served as the negative control. After 24 h of treatment, the cells were rinsed with PBS, fixed with 4% paraformaldehyde, and stained with 1 μg/mL of 4’6-diamidino-2-phenylindole (DAPI, Sigma Aldrich). Mounting medium (ibidi, Gräfelfing, Germany) was applied before imaging. Widefield imaging was conducted using a THUNDER Imager Live Cell (Leica Microsystems, Wetzlar, Germany) mounted on a Leica DMi8 microscope stand with a 63×/1.40 NA objective (HC PL APO CS2 63×/1.40 OIL UV). Fluorescence excitation was achieved using LED light sources at 395 nm for DAPI and 488 nm for tubulin–GFP. The camera (Leica DFC9000 GTC) operated in 2 × 2 binning mode, resulting in a pixel size of 206 nm. Image analysis was performed using ImageJ 1.54f software (National Institute of Health, Bethesda, MD, USA). The experimental methodology and microscopy techniques have been previously described [12].

### 2.9. Molecular Interaction with α- and β-Tubulins

In order to study the affinity of lupeol and quercetin with tubulins, molecular docking in the defined mode was performed with α1B, βI, and βIVb microtubules (PDB ID: 5N5N). The C and H chains of 5N5N were selected for docking, and three grid boxes in three different binding sites of vincristine, paclitaxel, and colchicine were used to compare the binding affinity of the two compounds with their receptor. Their grid box dimensions were 76 × 70 × 60 Å spacing, 0.375 Å at the grid-center, x = 48.907 Å, y = 37.217 Å, and z = 199.886 Å, 54 × 68 × 42 Å spacing, 0.375 Å at the grid-center, x = 49.973 Å, y = 32.388 Å, and z = 177.504 Å, and 126 × 100 × 124 Å spacing, 0.242 Å at the grid-center, x = 52.555 Å, y = 44.736 Å, and z = 169.256 Å, respectively. The molecular docking was performed with AutoDock 4.2.6, and the 3D and 2D pictures showing the interaction of the amino acids with the protein tubulin 5N5N were made using BIOVIA Discovery Studio Visualizer 2021. The mean and SD values were calculated with Microsoft Excel 2021 (Version 2306, Build 16.0.16529.20164) from three independent dockings.

### 2.10. Cell Cycle Analysis

U2OS cells (250 × 10^3^ cells/well) were seeded and treated for 72 h with a concentration of 1 × IC_50_ and 4 × IC_50_ of lupeol and quercetin, DMSO (negative control), and vincristine (positive control, 1 × IC_50_) (obtained from the University Hospital Pharmacy, Mainz, Germany). The cells were harvested and centrifuged with cold PBS twice (1500 rpm for 5 min). Cold ethanol (80%) was used for fixation. Samples were kept at −20 °C for 72 h. Before measurement, the cells were suspended with RNAs (Roche Diagnostics, Mannheim, Germany) and then incubated for 30 min. Lastly, 50 μg/mL of propidium iodide (PI) (Sigma-Aldrich) was added before the measurement. The DNA histogram was generated using FL2-A/histogram properties. All experiments were repeated three times independently. The cell cycle distributions were analyzed using FlowJo software (version 10.8.1) (Celeza, Olten, Switzerland) [41,42].

### 2.11. Cell Death Detection

CCRF-CEM cells (1 × 10^6^ cells/well) and U2OS cells (250 × 10^3^ cells/well) were seeded and treated for 72 h with varying concentrations of lupeol and quercetin. The treatment concentrations for CCRF-CEM cells included 1/4 × IC_50_, 1/2 × IC_50_, 1 × IC_50_, 2 × IC_50_, and 4 × IC_50_. For U2OS cells, the treatment concentrations were 1/2 × IC_50_, 1 × IC_50_, 2 × IC_50_, and 4 × IC_50_. DMSO was used as a negative control. A fluorescein isothiocyanate (FITC)-conjugated annexin V/propidium iodide (PI) assay kit (Bio version Biocat, Heidelberg, Germany) was used to detect apoptosis. The cells were washed with cold PBS and then with 1× binding buffer (Bio Version). Thereafter, 52.5 µL of annexin V master mix (2.5 µL of annexin V, 50 µL of 1 × binding buffer) was added to the cells and incubated at 4 °C in the dark for 15 min. Finally, 403 µL of PI master mix (3 µL of PI, 400 µL of 1 × binding buffer) was added to the cells. The experiments were performed three times independently [41,42].

### 2.12. Western Blotting

The HEK-Blue null1 cells were seeded for 24 h at a density of 500,000/well in 6-well plates and then treated for 24 h with 10 µM or 50 µM of β-amyrin, quercetin, and lupeol. DMSO served as a negative control. The next day, 100 ng/mL of TNFα was added and left for another 24 h. The cells were harvested on the 4th day, washed, and suspended with 1 mL of PBS × 2. The total protein was extracted using Mammalian Protein Extraction Reagent (M-PER) containing 1% protease inhibitor and phosphatase inhibitor (Thermo Fisher Scientifc, Darmstadt, Germany). The protein amounts were quantified using a microvolume spectrophotometer (NanoDrop, Thermo Fisher Scientific). Afterwards, 30 µg of protein extracts was loaded into each channel of 10% SDS-PAGE gel. After the separation and transfer steps, the polyvinylidene difluoride membrane was blocked in a TBST buffer consisting of 5% bovine serum albumin for 2 h. The membrane was incubated in 1:1000 primary antibodies rabbit mAb NF-κB p65 and GAPDH overnight and in secondary antibody 1:2000 anti-rabbit IgG HRP-linked for 1 h 30 (Cell Signaling Technology, Leiden, The Netherlands). All experiments were repeated three times. The mean and SD values were calculated with Microsoft Excel 2021 (Version 2306, Build 16.0.16529.20164) from three independent dockings. The significance level *p* value was calculated using *t*-test tails 2, type2.

### 2.13. Quantitative Real-Time RT-PCR

The RNA was extracted from the HEK-Blue Null 1 cells treated with 10 µM or 50 µM of β-amyrin, quercetin, and lupeol (Sigma–Aldrich) for 24 h and 100 ng/mL of TNF-α for 24 h. The extraction was performed with the RNeasy Kit from Qiagen (Hilden, Germany). The extracted RNA was converted to cDNA using Luna Script™ RT SuperMix Kit (E3010) following the instructions of the manufacturer (New England Biolabs GmbH, Frankfurt, Germany).

The primers for the *IL1B* and *IL6* genes (interleukin-1β and interleukin-6) were retrieved from the literature [43]. *GAPDH* primers were designed following the protocol previously reported by our team [44]. The primers were procured from Eurofins Genomics Germany GmbH (Ebersberg, Germany).

The real-time quantitative polymerase chain reaction (RT-qPCR) was carried out using 5× Hot Start Taq EvaGreen^®^ qPCR Mix (Axon-Labortechnik, Kaiserslautern, Germany) in a CFX384™ Real Time PCR Detection System (Bio-Rad Laboratories GmbH, Feldkirchen, Germany). The expression of the genes was normalized to *GAPDH*, and the fold of change was calculated with the 2^ΔΔCt^ method [45]. The experiments were independently repeated three times. The mean and SD values were calculated with Microsoft Excel 2021 (Version 2306, Build 16.0.16529.20164). The degree of significance was determined using the *t*-test tails 2 type 2 calculation method.

### 2.14. Statistical Analysis

For hierarchical cluster analysis of proteomic expression data, we used the method of Ward implemented in the WINStat program (Kalmia, CA, USA).

For survival analysis, we applied Kaplan–Meier statistics. The KM-Plotter database contains data from over 7000 samples across 21 different tumor types [46]. To assess the prognostic significance of *NFKB2* mRNA expression for cancer patient survival, we employed Kaplan–Meier statistics. We utilized false discovery rate (FDR) calculations to mitigate type I errors in multiple comparisons [47,48]. Specifically, we considered only Kaplan–Meier statistics with FDR rates of 5% or lower [49].

## 3. Results

### 3.1. Molecular Docking In Silico

Recently, we reported a chemical library of chamomile compounds [12]. In the present investigation, 212 chamomile compounds were utilized for molecular docking with the homodimer structure of NF-κB p65-RelA (PDB: 1NFI) by means of the AutoDock 4.2.6 program (Appendix A). Out of these 212 molecules, the top 28 were selected for further analysis. Table 1 shows their lowest binding energies (LBE, kcal/mol) as well as their predicted inhibition constants (pKi, µM).

The LBE and pKi values of these 28 compounds significantly correlated with each other using the Pearson correlation test (*p* = 2.61 × 10^−6^; *r* = 0.76; Figure 1A). Fifteen LBE values were below −6 kcal/mol, with a range between −8.70 ± <0.01 kcal/mol for β-amyrin and −6.01 ± 0.12 kcal/mol for chlorogenic acid. Their pKi values ranged between 0.42 ± <0.01 µM and 39.73 ± 7.82 µM, respectively (Table 1).

The top 6 of the 212 compounds were selected for further detailed studies. They were bound to three different pockets within two domains (Figure 1B). In the dimerization domain, daucosterol and apigenin shared the same pocket. On the other hand, β-amyrin, β-sitosterol, β-eudesmol, and triptolide were also bound to the same site, while myricetin was bound to the head of the N-terminal domain. Their interaction and binding with the protein were displayed as two- and three-dimensional figures (Figure 1C–H). Triptolide, as an established inhibitor of NF-κB, was used as a positive control [50]. Its interaction with NF-κB p65 RelA is visualized in Figure 1I.

### 3.2. Microscale Thermophoresis

To exemplarily verify the molecular docking results, we performed microscale thermophoresis (MST) with β-amyrin and NF-κB. Decreasing concentrations of β-amyrin were titrated against the human recombinant NF-κB. The working solutions of β-amyrin were obtained by diluting the stock solution in DMSO with working buffer (MST buffer). The equilibrium constants K_D_ confirmed that β-amyrin was indeed bound to NF-κB. By using the law of mass action, a K_D_ value of 943 ± 113 nM was determined (Figure 2A).

### 3.3. NF-κB Reporter Assay

To verify the results predicted in silico, these six phytochemicals were subjected to an NF-κB reporter cell assay. We measured the remaining NF-κB activity upon treatment with these compounds at concentrations of 0.1 µM, 1 µM, and 10 µM (Figure 2B). β-Amyrin was the most effective inhibitor of NF-κB activity, with remaining activity values of 38.9% ± 7.3, 52.2% ± 6.2, and 60.3% ± 8.7, respectively. Daucosterol had the lowest inhibitory activity with activity percentages of 6.1% ± 7 (0.1 µM), 16.2% ± 2.9 (1 µM), and 20.8% ± 2.7 (10 µM). Triptolide as a positive control strongly inhibited the NF-κB activity.

To see whether the rest activity measured in vitro may correlate with the LBE values determined in silico, we performed correlation analyses. The NF-κB rest activity (%) significantly correlated with the LBE values (kcal/mol) at 0.1 µM (*p* = 0.02; *r* = 0.89) and 1 µM (*p* = 0.03; *r* = 0.85). The treatment at 10 µM did not correlate with the LBE values *r* = −0.15 (Figure 2B–D).

### 3.4. Assessment of Oxidative Stress

It is well-known that high levels of reactive oxygen species (ROS) activate NF-κB and foster the inflammation process. In addition, NF-κB increases ROS generation during inflammation [51,52]. Therefore, we measured the ROS levels as a parameter of oxidative stress. Untreated HEK-Blue Null 1 (HBN1) cells exhibited very low ROS levels (0.68%). However, if the cells were exposed to hydrogen peroxide (H_2_O_2_) as the positive control, the ROS generation significantly increased by 11.63% (*p* = 0.02, compared with untreated cells as the negative control). The combination of H_2_O_2_ and tumor necrosis factor α (TNF-α) as a proinflammatory cytokine further increased the ROS generation up to 14.13% (*p* = 0.005, compared with the negative control). In contrast, β-amyrin treatment significantly decreased the H_2_O_2_- and TNF-α-induced ROS generation by 12.30% (*p* = 0.1), 10.83% (*p* = 0.05), and 8.80% (*p* = 0.01) at 0.1 µM, 1 µM, and 10 µM, respectively (Figure 3).

### 3.5. Measurement of the Mitochondrial Membrane Potential

Mitochondria are not only essential for ATP generation and programmed cell death but also play a key role in inflammation [53]. We investigated whether the inhibition of NF-κB after TNF-α induction affected the mitochondrial membrane potential and whether β-amyrin reversed this effect. HEK-Blue Null1 cells were treated with 10 µM of β-amyrin for 24 h and with TNF-α for 3 h. As a positive control, cells were treated with 10 µM of vinblastine (Figure 4). Using flow cytometry and JC-1, we measured a considerable breakdown of the mitochondrial membrane potential (MMP) upon treatment with β-amyrin or vinblastine compared to the untreated control (Figure 4A). β-Amyrin led to a 52.6% increase in the population of cells with disrupted mitochondrial membrane potential (indicative of apoptotic or dead cells), while only 47.6% of cells maintained their MMP (and their viability). In comparison, vinblastine had an effect of 73.6% living cells and 26.6% dead cells (Figure 4B).

### 3.6. Cytotoxicity and Oncobiogram Analyses

The link between inflammation and carcinogenesis is well-established [54]. Accordingly, the first part of the analysis was to compare the effect of the multiple chamomile compounds on the cell viability of 60 cell lines originating from different tumor types. While (+)-catechin, β-sitosterol, daucosterol, caffeic acid, and β-eudesmol were inactive or only minimally active, the strongest inhibition was observed with apigenin and farnesol. Lupeol and quercetin showed intermediate cytotoxicity (Figure 5A–C). Because apigenin and farnesol as the most cytotoxic compounds in this panel were already the subjects of previous investigations by us [19,20,55,56,57], we chose quercetin and lupeol for further analyses.

As β-amyrin was not included in the NCI database, we performed a growth inhibition assay, which revealed that β-amyrin did not exert any cytotoxic effect on the sensible leukemia cells. The highest inhibition was observed at a concentration of 100 µM (26.2% ± 9.6) (Appendix A).

Then, we compiled the sensitivity of the NCI tumor cell lines towards lupeol and quercetin according to their origin, i.e., leukemia, melanoma, and brain tumors as well as carcinoma of the colon, ovary, breast, kidney, lung, and prostate. The mean log_10_IC_50_ (M) values for quercetin and lupeol were compared with those for chlorambucil as an established anticancer drug (control). It is evident from the data shown in Figure 5D that all three compounds exhibited the greatest growth-inhibitory efficacy in leukemia cells compared to the cell lines of other tumor types. Melanoma cells were most resistant to quercetin. With some variations, the cytotoxicity of both lupeol and quercetin was approximately comparable to that of chlorambucil.

While natural products are bioactive through multiple mechanisms [58,59], the exact modes of action of lupeol and quercetin are unknown. Therefore, we correlated the log_10_IC_50_ values of lupeol and quercetin with those of 91 standard agents with known modes of action (Figure 5E). Both lupeol and quercetin were most frequently correlated with microtubule inhibitors and mTOR inhibitors. Therefore, we subjected both compounds to oncobiogram analysis by compiling the correlation coefficients (*r*-values) of lupeol and quercetin to those of 10 known microtubule inhibitors (Figure 5F). The analysis favored the hypothesis that lupeol and quercetin might also act as tubulin inhibitors.

### 3.7. Inhibition of α-Tubulin by Lupeol and Quercetin as Detected through Confocal Immunofluorescence Microscopy

The oncobiogram analysis showed that both lupeol and quercetin might act as microtubule inhibitors. To validate this hypothesis, we treated the U2OS cells transfected with GFP-α-tubulin with lupeol and quercetin (0.1 µM, 1 µM, and 10 µM). Vincristine (1 µM) was used as the control drug to monitor the inhibition of microtubule polymerization, and paclitaxel (1 µM) was used as the control drug to monitor the inhibition of microtubule depolymerization. As can be seen in Figure 6, both lupeol and quercetin affected α-tubulin in a comparable manner to vincristine, indicating that they may have inhibited microtubule polymerization rather than depolymerization.

### 3.8. Binding of Lupeol and Quercetin to α-Tubulin as Detected through Molecular Docking

To further investigate the possible interactions of lupeol and quercetin to *α*-tubulin, we performed molecular docking. For quality control of the results obtained in this series of molecular dockings, we first correlated the LBE and pKi values. The Pearson correlation test showed that all LBE values (kcal/mol) correlated significantly with their respective pKi (µM) values at *p* = 0.004 and *r* = 0.85 (Figure 7A).

The in silico analysis demonstrated that lupeol and quercetin had higher binding affinities to α-tubulin at the vincristine binding site compared to the binding sites of paclitaxel and colchicine (Figure 7A). Specifically, the calculation of low binding energy (LBE) for lupeol interacting with α-tubulin at the vincristine’s binding site was lower than its LBE value at paclitaxel’s binding site, with values of −8.62 ± 0.04 kcal/mol and −7.12 ± 0.01 kcal/mol, respectively. Similarly, quercetin had an LBE with the protein at vincristine’s binding site compared to paclitaxel’s binding site, with values of −6.77 ± 0.06 kcal/mol and −5.99 ± 0.29 kcal/mol, respectively. In contrast, both compounds had less affinity with the α-tubulin if docked at the colchicine’s binding site, with calculated LBE values of −4.48 ± 0.08 kcal/mol for lupeol and −4.72 ± 0.15 kcal/mol for quercetin (Table 2). Figure 7C–E show the 3D and 2D docking poses of vincristine, quercetin, and lupeol, highlighting the amino acid residues involved in the binding of each molecule to α-tubulin at the vincristine binding site.

### 3.9. Cell Cycle Analysis

Both quercetin and lupeol significantly induced G2/M phase arrest in U2OS cells at 4 × IC_50_) after 72 h compared to the negative control (DMSO) (Figure 8). Quercetin induced cell cycle arrest with 86.6% of the cells in the G2/M phase (*p* = 0.0001) and 4.9% in the S phase (*p* = 0.4). Interestingly, a small but significant portion of cells (7.9%) was also increased in the G0/G1 phase (*p* = 0.0001), indicating that arresting the cells in the G2/M phase was not the only consequence of treatment. As expected, the positive control vincristine (1 × IC_50_) induced cell cycle arrest, with 83.6% of cells in the G2/M phase (*p* = 0.001), 5.9% in the S phase (*p* = 0.2), and 9.3% in the G0/G1 phase (*p* = 0.001).

Similarly, lupeol also induced cell cycle arrest in U2OS cells, with 90.9% of the cells in the G2/M phase (*p* = 0.0002), 5.3% in the S phase (*p* = 0.05), and 3.8% in the G0/G1 phase (*p* = 0.0001). Vincristine was used in parallel as the positive control in this experiment, too. A concentration of 1 × IC_50_ induced cell cycle arrest, with 82.3% of cells in the G2/M phase (*p* = 0.002), 6.0% in the S phase (*p* = 0.02), and 12.3% in the G0/G1 phase (*p* = 0.001).

### 3.10. Drug Resistance Profiling of Lupeol and Quercetin

The question arises as to whether lupeol and quercetin are recognized by classical drug resistance mechanisms and, therefore, whether their activity is hampered by these drug resistance mechanisms. Utilizing the NCI database [40], we correlated the log_10_IC_50_ values of lupeol and quercetin with various mechanisms of multidrug resistance in the NCI panel of tumor cell lines (Table 3). Notably, we did not observe an involvement of the two compounds in any drug resistance phenotype, except for one correlation between quercetin and *ABCB5* mRNA expression analyzed through qPCR. This correlation was not significant if mRNA expression was analyzed through microarray hybridization. These findings indicate that these two compounds from chamomile might be useful in inhibiting tumor cells that are otherwise resistant to drugs.

### 3.11. Cytotoxicity Assays for Lupeol and Quercetin

Because the analysis of the NCI cell line panel indicated the activity of both lupeol and quercetin against classical drug resistance mechanisms, we performed resazurin assays to investigate whether these results obtained through correlation analyses could be verified by independent in vitro experimentation.

The dose–response curves in Figure 9 show that multidrug-resistant P-glycoprotein-overexpressing CEM/ADR5000 leukemia cells were cross-resistant to quercetin (degree of resistance: >7.3-fold) and lupeol (degree of resistance: 3.2-fold). The glioblastoma transfected cells with ΔEGFR (U87.MG/ΔEGFR) were 2.3-fold resistant to quercetin and 7.4-fold resistant to lupeol compared to wild-type U87.MG cells. Colorectal *TP53* knockout cells (HCT116 p53^−/−^) were only weakly resistant to quercetin and lupeol (1.4-fold and 1.8-fold, respectively) compared to their corresponding wild-type cells (HCT116 p53^+/+^). All degrees of resistance were, thus, rather low, indicating no or only very low levels of resistance to lupeol and quercetin. These results fit the data obtained from the NCI tumor cell line panel, showing no correlations between the log_10_IC_50_ values and the classical resistance mechanisms. The IC_50_ values of U2OS for quercetin and lupeol were higher than those found for most other cell lines tested, indicating that these osteosarcoma cells were more resistant to these two compounds.

### 3.12. Cell Death Detection

We investigated the mode of cell death in CCRF-CEM cells upon treatment with lupeol and quercetin for 72 h. The FITC-conjugated annexin V/PI assay was used to distinguish between living, early apoptotic, late apoptotic/necrotic, and primary necrotic cells. Annexin V is usually detected in early and late apoptosis. However, PI staining detects cells in late apoptosis and necrosis. Figure 10 shows that both compounds significantly induced cell death compared to the negative control. Quercetin induced late apoptosis/necrosis in 58.70% of cells at 4 × IC50 (*p* = 0.002), while lupeol induced late apoptosis/necrosis in 73.80% of cells (*p* = 0.001).

### 3.13. Western Blotting

The NF-κB reporter assay in Figure 2 indicated that the selected chamomile compounds inhibited the activity of the NF-κB promoter in the context of anti-inflammatory NF-κB activity. As NF-κB inhibition also plays a role in the apoptosis of cancer cells, we performed a Western blotting experiment to examine the effect of three chamomile compounds (β-amyrin, lupeol, and quercetin) on the protein expression level of NF-κB 65. β-Amyrin and quercetin (50 µM) increased cytoplasmic NF-κB expression, while lupeol displayed non-significant tendencies (Figure 11A).

### 3.14. Quantitative Real-Time RT-PCR

*IL-1β* and *IL-6* are known downstream cytokines regulated by NF-κB that are important for mediating proinflammatory and cell-proliferative signals. All three compounds significantly downregulated the expression of the genes encoding *IL-1β* and *IL-6*, particularly at 50 µM (Figure 11C). Notably, quercetin was significantly more effective in downregulating the two genes compared to the other two compounds, with a fold-change in log_2_ of −1.11 for *IL-6* (*p* = 0.01) and −0.71 for *IL-1β* (*p* = 0.01). Conversely, β-amyrin displayed the lowest downregulatory activity for IL-6, with fold-changes of log_2_ −0.29 for *IL-6* and −0.56 for *IL-1β* (*p* = 0.01). Furthermore, lupeol decreased the genes with log_2_ values of −0.54 for *IL-6* (*p* = 0.01) and −0.65 for *IL-1β* (*p* = 0.02).

### 3.15. Proteome Analysis

If all or most of the classical drug resistance mechanisms are not operative for lupeol and quercetin, the question arises as to which factors do determine sensitivity and resistance to these two drugs. Therefore, we conducted a COMPARE analysis by correlating the expression levels of 3171 proteins in the 60 tumor cell lines from the NCI database [40] with the log_10_IC_50_ values of both lupeol and quercetin. Subsequently, the top 40 proteins were assembled, and 20 were directly correlated with the two compounds and 20 were reversely correlated. Finally, we generated a two-dimensional clustering and color-coded heat map using a hierarchical Ward cluster method, where, in one dimension, the expression profiles of the 40 proteins appear and, in a second dimension, the different cell lines appear (Figure 12 and Figure 13). The log_10_IC_50_ values for lupeol and quercetin, which were not included in the cluster calculations, are shown on the right side of the heat maps for illustration. The cellular responsiveness to both lupeol and quercetin was determined by comparing individual log_10_IC_50_ values to the median value across all cell lines. If the log_10_IC_50_ value was lower than the median, the cell lines were categorized as sensitive; if it was above the median, the cells were defined as resistant.

For both lupeol and quercetin, we obtained three clusters (clusters 1–3) showing the individual protein expression. Furthermore, we obtained two clusters of cell lines (clusters A and B) (Figure 12 and Figure 13). These clusters were then correlated with the log_10_IC_50_ values (which were not included in the cluster calculation). Interestingly, clusters 1 and 2 predominantly included lupeol/quercetin-resistant cell lines, while cluster 3 predominantly contained sensitive cell lines. Cluster A predominantly contained cell lines with high protein expression in cluster 1 and low expression in cluster 3, while in cluster B, the protein expression was the opposite (low in cluster 1 and high in cluster 3). Afterwards, we calculated the difference between sensitive and resistant cell lines using the χ^2^ test. The results showed that the distribution of the three clusters was statistically significant for both lupeol (*p* = 1.44 × 10^−4)^ and quercetin (*p* = 0.044) (Figure 12 and Figure 13).

Each of the 40 proteins was identified through this approach to determine the sensitivity or resistance to lupeol or quercetin. These proteins have been compiled along with their protein symbols, full names, cellular functions, and functional categories in Appendix A. Although these proteins seemed to be unrelated at first sight, it was interesting to observe that they belonged to some common functional categories. These categories are shown in Table 4.

### 3.16. Kaplan–Meier Survival Analysis

Because NF-κB is known for its anti-proliferative, anti-apoptotic, and anti-inflammatory activities in the cell [60], we wanted to survey the role of NF-κB expression for the survival prognosis of cancer patients. We performed Kaplan–Meier survival analysis with the help of the KM-Plotter database. Out of 21 cancer types, the analysis revealed that the higher the *NFKB2* mRNA expressed, the lower the survival time for patients with renal clear cell carcinoma compared with patients with low *NFKB2* expression. Moreover, patients with high *NFKB2* expression in grade 3 renal carcinoma died significantly earlier than those with a low expression. Interestingly, the female patients with renal carcinoma had significantly longer survival times compared with males if *NFKB2* mRNA expression was low. Similarly, patients with low neoantigen load and low *NFKB2* mRNA expression in their renal carcinoma cell type had a better survival prognosis than those with high *NFKB2* expression and a significantly higher survival time than the patients with a low mutational rate (Figure 14). These data indicated that the inhibition of NF-κB could indeed be beneficial in the prevention and treatment of renal carcinoma.

## 4. Discussion

Chamomile tea is not only used for its exquisite taste but also its numerous health-promoting effects. Renowned for centuries, chamomile contains metabolites that possess anti-inflammatory, antioxidant, and relaxant properties [61,62]. Over decades, many studies have revealed the anti-inflammatory effect of chamomile both in vitro and in vivo [63]. They have demonstrated the inhibitory effect of chamomile extract against markers and pathways at the molecular and cellular levels [64,65]. These findings were complemented by in vivo research, where chamomile compounds exhibited anti-inflammatory effects if tested in animals [66,67]. Adding to this, various clinical studies have shown evidence of the anti-inflammatory effect of chamomile if tested in patients with different types of inflammatory disease [65,68,69] Moreover, chamomile has garnered attention for its potential in cancer prevention and treatment [70,71,72]. In vitro results have suggested that chamomile extracts may possess anticancer properties, showing inhibitory effects on the proliferation of cancer cells and in some cases on programed cell death [12,13,73,74,75,76,77]. While there is limited evidence in vivo, some preliminary evidence supports the potential of chamomile as an anticancer and preventive agent [78,79,80]. The entangled connection between inflammation and carcinogenesis may make a reasonable contribution to this point of view [81,82].

In the same entangled manner, the transcription factor NF-κB was proven to play a role in both inflammation and carcinogenesis [83]. NF-κB not only controls the inflammatory responses with its transcriptional activity on several proinflammatory promoters [84] but also has a crucial role in regulating apoptosis and cell proliferation, mostly by favoring cell survival mechanisms and, therefore, contributing to the initiation and progression of tumors [85,86]. Additionally, NF-κB pathways also play a pivotal role in the development of resistance against anticancer drugs [87,88]. Interestingly, the development of tumors is often driven by similar mechanisms that also cause drug resistance. It is in the nature of the biological system of all organisms to detoxify and excrete harmful substances, such as toxic metabolic products used in chemotherapy, through the activation of pro-survival pathways. This will then annul the induction of cell death and enhance the continuous progression of the tumor in the body [89,90,91].

These findings and observations prompted the question of whether chamomile or its derivates inhibit NF-κB. NF-κB activation or inhibition plays a key role in cellular processes, such as cell survival, cell growth, cell proliferation, and even cell death in stem cells and different cancer cells, including leukemia [92,93,94,95]. To address this, we initiated virtual drug screening to investigate the bindings between 212 chamomile compounds to NF-κB. Subsequently, 28 compounds were further selected for a computational assay in which we calculated the LBE and pKi values. Among these 28 molecules, 15 exhibited LBE values consistently below −6 kcal/mol. Correspondingly, the pKi values demonstrated a diverse range, fluctuating between 0.42 ± <0.01 µM for β-amyrin and 46.78 ± 0.12 µM for apigenin. Six compounds were selected from these results for further in vitro experiments. β-Amyrin, β-sitosterol, β-eudesmol, daucosterol, and myricetin were chosen depending on their binding energy, including a range from the highest to the average and lowest values. Apigenin, with higher binding, was chosen for its popularity and proper comparison with the other compounds. Using an inhibitory NF-κB reporter cell assay where the rest activity of NF-κB was measured, β-amyrin exhibited the highest percentage of inhibition. In contrast, daucosterol demonstrated the lowest inhibitory activity. These experimental results confirm our in silico predictions. Hence, the regression analysis of the data showed that NF-κB rest activity (%) significantly correlated with the LBE values. Additionally, we validated the authenticity of our data using microscale thermophoresis (MST). This result further confirmed the binding of β-amyrin to NF-κB.

It is now evident that activated NF-κB enhances the expression of proinflammatory genes in a positive feedback loop, leading to increased ROS generation in the cell. Elevated ROS levels during chronic inflammatory diseases or cancer can in return further activate NF-κB and upregulate immune responses [96,97]. Accordingly, to validate this relationship, we performed an ROS generation assay using HEK-Blue Null 1 cells treated with β-amyrin and exposed to 100 ng/mL of TNF-α and 10 µM of H_2_O_2_, thus creating a high oxidative microenvironment within the cell. The results demonstrated that β-amyrin reduced ROS levels significantly. Moreover, it is well-established that both acute and chronic inflammatory diseases are characterized by dysfunction of mitochondria and aggregation of ROS generation. These factors can lead to cell damage or cell death. Alternatively, in other cases, they can trigger the activation of cell survival pathways and cell proliferation [98,99]. In this context, we examined whether the inhibition of NF-κB with β-amyrin might affect the mitochondrial membrane potential (MMP). Employing the MMP-JC1 assay on HEK-Blue Null 1 treated with β-amyrin and TNF-α (to stimulate NF-κB activation), we observed a significant impact on the breakdown of MMP in the treated samples compared to the control. β-Amyrin treatment increased the population of MMP-disrupted cells by 52.6%. In comparison to vinblastine, 26.6% of the cell population were unhealthy or dead. These results highlight the promising potential of β-amyrin as a potential therapeutic agent for inflammatory diseases and oxidative-stress-related disorders, suggesting its applicability as a cancer-preventive agent or even an adjunctive treatment in cancer therapy. 

This leads to the second part of this study, where we studied the therapeutic value of chamomile, particularly focusing on inflammation, cancer prevention, and overcoming drug resistance. We started with cytotoxicity and oncobiogram analyses comparing the effects of multiple chamomile compounds on the cell viability of 60 cell lines originating from various tumor types using the NCI60 database. Notably, quercetin and lupeol exerted significant cytotoxicity, with specific preferences against colon cancer for lupeol and melanoma for quercetin. The oncobiogram analysis further revealed correlations with microtubules and mTOR inhibitors, validating previous studies in which both quercetin and lupeol disturbed the polymerization of microtubules through their binding to tubulin [100,101] and inhibited the mTOR pathway [102,103]. The immunofluorescence microscopy experiment validated these findings, showing clear inhibition of the GFP-α tubulin polarization by lupeol and quercetin (at 10 µM) comparable to the effect of vincristine (at 1 µM). Given the key role of microtubules in cell stabilization and maintenance, the identification of novel tubulin inhibitors for tumor therapy remains essential [104]. Furthermore, the in silico assessment confirmed that both lupeol and quercetin have higher binding affinities with α-tubulin if docked at vincristine’s binding site compared to the two other known tubulin inhibitors, paclitaxel and colchicine.

It is well-established that paclitaxel enhances the stability of the microtubules, maintaining them in their polymerized form, which induces the activity of NF-κB through the same pathway as TNF-α. In contrast, colchicine and vincristine are known to destabilize and disturb the microtubule formation, which blocks the translocation of the cytoplasmic NF-κB p65 to the nucleus and reduces the activity of NF-κB through the TNF-α canonical pathway [105,106,107]. We hypothesize that lupeol and quercetin may act in the same manner as vincristine by inhibiting the microtubules and reducing the activity of NF-κB.

Microtubules are also important for immune reactions and inflammatory processes, as microtubule networks are involved in the migration of immune cells to inflamed tissues [108,109,110]. Thus, the inhibition of microtubules by anti-inflammatory compounds from chamomile might not only directly inhibit tumor cells but also suppress the activation and migration of inflammatory cells and thereby contribute to the decline of the inflammation process [111].

There is not only a connection between NF-κB and the microtubule but also cytokines. The NF-κB-mediated release of proinflammatory cytokines (e.g., *IL-1β* and *IL-6*) favors not only the activation of immune cells at inflamed sites in the body but also cell survival and cell proliferation. Because receptors for proinflammatory interleukins are expressed at both immune cells and tumor cells, their inhibition by natural products contributes to the suppression of tumor growth [112,113,114]. We found that β-amyrin, lupeol, and quercetin inhibited the expression of *IL-1β* and *IL-6*, a result that is consistent with the point of view that inhibition of interleukin production also causes the inhibition of tumor cell growth. At the same time, these three substances increased the expression of NF-κB of HEK-Blue Null1 cells. This can be explained by an inhibited translocation of NF-κB from the cytosol to the nucleus, which consequently leads to suppressed interleukin production due to the transcriptional activity of NF-κB in the nucleus. Because microtubules are involved in the translocation of NF-κB from the cytosol to the nucleus [115,116,117], microtubules inhibited by chamomile compounds may be unable to translocate cytosolic NF-κB to the nucleus, where it can bind to promoters of NF-κB downstream genes. Chamomile compounds may therefore act on multiple intracellular targets and pathways.

Recognizing that chemoresistance represents a major obstacle in cancer treatment, we conducted drug resistance profiling for quercetin and lupeol by correlating their values to classical drug resistance genes, including ABC transporters, oncogenes, tumor suppressors, and other mechanisms. The analysis indicated that the two compounds have no cross-resistance and are not involved in classical drug resistance mechanisms. This is a remarkable result as compounds combatting or bypassing resistance to clinically established drugs are urgently required.

We aimed to verify the correlation analyses with the NCI panel by using specific cell models that showed specifically overexpressed or knocked-out genes conferring drug resistance (P-glycoprotein, *EGFR*, *TP53*). While it is known that established cancer drugs can provoke high levels of resistance in tumors (e.g., CEM/ADR5000 cells were more than 1000-fold resistant to doxorubicin) [118], lupeol and quercetin showed no or only very low degrees of resistance of three or less. These results confirmed our bioinformatic analysis, indicating that both compounds are active against otherwise drug-resistant tumor cells and do not encounter major cross-resistance mechanisms.

It is reasonable to propose that chamomile holds not only the potential to prevent inflammation and cancer but also the ability to overcome drug resistance. This theorizes a dual role for chamomile as a preventive measure but also as an additive to cancer treatment, particularly in cells that exhibit resistance to conventional therapies. These insights open new possibilities for exploring chamomile compounds, such as β-amyrin, lupeol, and quercetin, as valuable assets in addressing the challenges posed by drug-resistant cancer cells. Further research and clinical studies could elucidate the full scope of chamomile’s contributions to cancer prevention and treatment.

In this context, it was interesting that the cytotoxicity of both compounds in CCRF-CEM resulted in late apoptosis/necrosis. This might be a cell-type-specific effect, as quercetin and lupeol have been reported to induce either apoptosis or necrosis (or necroptosis) in different cell lines [119,120,121,122,123,124]. Because apoptosis is driven by the balance of pro- and anti-apoptotic proteins in specific signaling cascades, it is known that mutations in specific genes encoding these proteins confer resistance to apoptosis [125,126,127,128]. As a consequence, cytotoxic insults may overcome apoptosis resistance by other cell death modes [129].

Despite the fact that classical drug resistance mechanisms did not affect the responsiveness to lupeol and quercetin, it can be assumed that the different responsiveness of the NCI cell lines may be due to variations in the expression of genes that influence sensitivity and resistance to lupeol and quercetin. Therefore, the question arises as to what are the cellular factors that determine sensitivity or resistance to these two compounds. To explore this further, we analyzed the proteomic expression of the NCI tumor cell lines. In this analysis, we identified 40 proteins whose expression significantly correlated with sensitivity and resistance of the cell lines. Although the two sets of each of the 40 proteins identified through hierarchical cluster analysis for lupeol and quercetin seem to be heterogeneous at first sight, they belong to some common functional categories, i.e., protein and vesicle trafficking, DNA/RNA metabolism, immune function, cytoskeleton, mitochondrial function, cell proliferation and differentiation, cell death, channels and transporters, chaperone functions, signal transduction, and other functions (Table 4). Although these proteins have not been described or only scarcely described so far to determine sensitivity or resistance to lupeol or quercetin and the relationships found in our study are novel, most of the functional protein groups mentioned above are known to influence the response of tumor cells to standard chemotherapy. Therefore, it can be hypothesized that these proteins might also play a role for quercetin and lupeol.

*Protein and Vesicle Trafficking*: The translocation of proteins and vesicles to their functional sites in tumor cells has received less attention in the context of drug resistance. Impaired distribution of both cellular structures may, however, indeed influence cellular homeostasis and, therefore, contribute to carcinogenesis and differential responses to chemotherapy [130,131]. Our data obtained with lupeol, and quercetin indicate that there is still an open field for innovative research to uncover the full range of trafficking mechanisms for responses to established cancer drugs as well as cytotoxic natural products.

*DNA/RNA Metabolism*: This is a huge field spanning many topics, such as transcription and transcription factors [132], translation [133], DNA replication [134], chromatin remodeling [135], chromosome segregation [136,137], etc. These processes are central to the vital functions of tumor cells and play important roles under the selection pressure of anticancer therapies. Understanding these mechanisms in more detail will facilitate the development of novel targeted therapies to address these mechanisms in cancer cells.

*Immune Functions:* Although tumor immunology has been a thriving field in cancer research, attention has recently focused more on the advent of passive immunotherapies (e.g., therapeutic monoclonal antibodies, immune checkpoint inhibitors) and active immunotherapeutic approaches (e.g., CAR-T). On the other hand, it is also known that the immune system influences the responsiveness of tumors to classical chemotherapy by generating therapy-resistant niches in tumors [138].

*Cytoskeleton:* Among the cytoskeletal proteins, not only the microtubules as direct targets for Vinca alkaloids, taxanes, and new drug candidates are well-known [139] but also others, such as actin, vimentin, and cytokeratin as well as proteins accompanied by cytoskeletal proteins that contribute to cytoskeletal organization and basic tumor cell features (e.g., cell remodeling during invasion and metastasis, resistance to cisplatin and other drugs, etc.) [140,141,142].

*Mitochondrial Functions:* Mitochondria do not only supply cancer cells with energy (in terms of ATP production) but also help them to adapt to cellular stressors, such as nutritional deprivation hypoxia and oxidative stress. Mitochondria provide sufficient biosynthetic flexibility to tumors to survive even under worse conditions [143,144]. Therefore, mitochondrial metabolism is involved in tumor progression and drug resistance [145].

*Cell Proliferation and Differentiation:* It has been long recognized that slowly growing tumors are more resistant to chemotherapy than fast-growing ones [146]. As a consequence, the cellular differentiation state, e.g., driven by the epidermal growth factor receptor (EGFR), also influences drug response [147,148]. The proteomic analyses in the present study (Table 4) indicate that several proteins involved in cellular differentiation determine sensitivity or resistance to cytotoxic natural products.

*Cell Death:* Apoptosis as most well-known mode programmed cell death is long known to affect the efficacy of chemotherapy and most other cytotoxic compounds as well [149]. However, in recent years a surprising number of non-apoptotic modes of programmed cell death have been uncovered [150]. Interestingly, these newer forms of cell death also influence the sensitivity and resistance of tumors to chemotherapy [151,152]. In the present investigation, we found that lupeol and quercetin induced necrosis rather than apoptosis. In our proteomic analyses, we also found that several proteins involved in apoptotic and non-apoptotic cell death correlated with these two natural products (Table 4).

*Ion Channels and Drug Transporters:* The membrane as the first barrier for cytotoxic compounds to enter cells represents an obvious cause of drug resistance. Though transporter molecules from the ATP-binding cassette (ABC) and solute carrier (SLC) families have been intensively studied for their role in cancer drug resistance [153,154], other membrane proteins (e.g., chloride and other ion channels) are associated with cell growth and apoptosis and, thereby, affect the response of tumor cells to chemotherapy [155]. In our own investigations during a period of more than two decades, we found that natural products are a rich source serving either as substrates and inhibitors of drug transporters and even as a new strategy to exploit hypersensitivity (collateral sensitivity) to treat otherwise drug-resistant cells [156]. Our present analysis demonstrated that channels and transporters are also involved in their responsiveness of tumor cells to natural products, such as lupeol and quercetin (Table 4).

*Cell Adhesion:* It is well known that single-cell tumors, such as leukemia, tend to respond better than solid cancer. One reason may be that solid cancers grow in tissues where the cells are connected to and communicating with each other numerous proteins including cell adhesion molecules, integrins, adapter proteins, and associated kinases. This set of proteins has been termed “cell adhesion resistome” because these proteins contribute to anticancer drug resistance [157]. The present proteomic analyses revealed a number of proteins also belonging to this cell adhesion resistome (Table 4).

*Chaperone Functions*: Molecular chaperones are responsible for the correct folding of proteins either in the nascent state if freshly translated amino acid chains are folded to three-dimensional protein structures or if cellular stress causes misfolding and damage of proteins. These quality control mechanisms are crucial for the correct function of proteins. Chaperones, i.e., heat shock proteins, are associated with drug resistance of tumors [158,159,160]. We found a couple of proteins also belonging to the chaperone system to be linked with response to lupeol and quercetin (Table 4).

*Signal Transduction:* A quantity of proteins involved in signal transductions that seems to be barely manageable contribute to complex signaling networks that drive cancer cells sensitive or resistant to cancer drugs and cytotoxic natural products [161,162,163]. Rather than single signaling pathways, sophisticated cross-talks between different signaling routes led to broad networks determining drug response [164,165]. The new discipline of network pharmacology is just beginning to understand these complex mechanisms. This is not also true for established cancer drugs but also for natural products [166]. It will be thriving to further detect single elements in large signaling networks that are responsible for drug response of tumor cells to lupeol and quercetin in the future.

*Other Functions:* Among the proteins that do not belong to the functional categories mentioned above are DNA repair proteins, tumor suppressors, and others. Because DNA repair mechanisms as well as tumor suppressors (e.g., TP53), are well-known drug resistance mechanisms [167,168], it is also worth speculating that the proteins identified in our analyses contribute to resistance to lupeol and quercetin. It is quite unexpected to find proteins involved in melanosome biogenesis and function among the proteins correlating with quercetin responsiveness. However, because melanosomes generate ROS [169], it is possible that there is a direct or indirect link to the response of anticancer drugs Further experiments are warranted to substantiate these findings.

In the present study, we investigated different mechanisms and targets. It became apparent that inflammation and tumor-killing properties of chamomile compounds are determined by multiple rather than by single mechanisms. The multi-specific action of chamomile and numerous other medicinal plants apparently represents a selection advantage during the evolution of plants on this globe. A large battery of defense mechanisms is superior to combat hostile organisms, such as microbes and herbivores. This principle is also realized in multi-compound pharmacology of medicinal plants. The phytotherapy relies on addressing multiple targets at the same time [58].

In the third part of the present investigation, we therefore focused on the role of NF-κB as prognostic factor for patient survival. Related to this topic the question of NF-κB as potential treatment targets for chamomile compounds addressed in the first two parts of the present study. May chamomile compounds be beneficial to prevent the onset of cancer or bypass drug resistance as additives to standard chemotherapy by inhibition of NF-κB? To clarify the prognostic relevance of NF-κB for survival of cancer patients we took advantage of the transcriptomic analyses of more than 7000 tumor biopsies derived from 21 tumor types of The Cancer Genome Atlas (TCGA) deposited at the KM-Plotter database (https://kmplot.com/analysis/ (accessed on 20 November 2023)). We observed that renal clear cell carcinomas with low NF-κB expression from different patho-clinical categories (gender, grade 3, low mutation status, low neoantigen rate) were significantly correlated with longer patient survival times than tumors with high NF-κB expression using Kaplan-Meier statistics. These results indicate that these subgroups of tumors might benefit from NF-κB inhibition by chamomile compounds.

## 5. Conclusions

In conclusion, this study highlights the significant therapeutic potential of chamomile compounds, specifically β-amyrin, lupeol, and quercetin, in addressing inflammation and cancer. Through a thorough investigation encompassing virtual screening, experimental assays, and proteomic analyses, we have demonstrated the inhibitory effects of these compounds on NF-κB signaling pathways and cytokine expression. Notably, our findings suggest that chamomile derivatives could serve as promising candidates for both preventive and adjunctive cancer therapies, particularly in overcoming drug resistance mechanisms. Moving forward, further research exploring the activity of the three compounds across all inflammation pathways and conducting clinical trials are essential. This would be beneficial to fully elucidate the clinical implications and therapeutic mechanisms of chamomile compounds, paving the way for the development of novel cancer treatments and personalized medicine approaches.

## Figures and Tables

**Figure 1 biomedicines-12-01484-f001:**
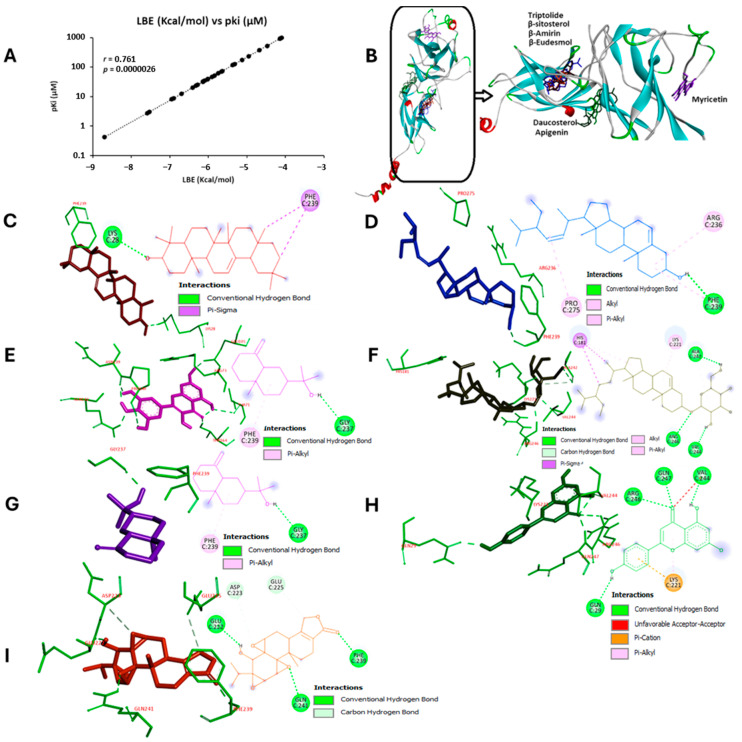
In silico binding of selected phytochemicals extracted from chamomile (*Matricaria chamomilla*) and triptolide (positive control) to NF-κB. Molecular docking analyses have been performed with NF-κB -RelA (PDB ID: 1NFI). (**A**) The lowest binding energies (LBE, kcal/mol) of the top 28/212 compounds (=10.4%) significantly correlated with the predicted inhibition constants (pKi, µM) (*p* = 2.61 × 10^−6^; *r* = 0.76). (**B**) The top 6/212 compounds were bound to different pockets within two domains. The interactions of these six compounds with the amino acids of NF-κB are displayed as 2D and 3D figures: (**C**) β-amyrin, (**D**) β-sitosterol, (**E**) myricetin, (**F**) daucosterol, (**G**) β-eudesmol, (**H**) apigenin, and (**I**) triptolide (positive control).

**Figure 2 biomedicines-12-01484-f002:**
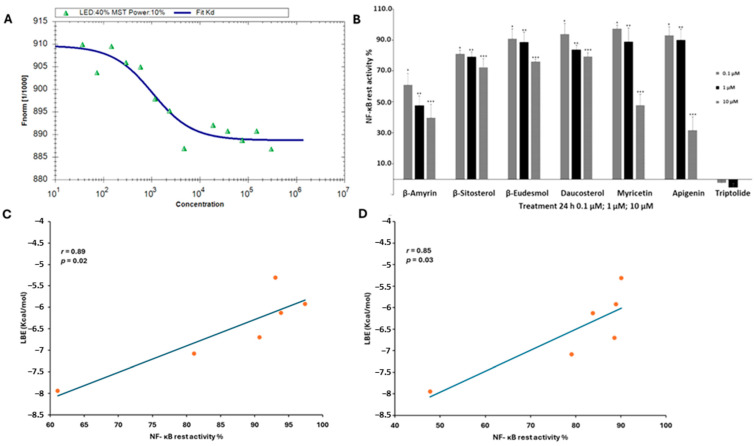
In vitro binding to and inhibition of NF-κB for selected phytochemicals extracted from chamomile (*Matricaria chamomilla*) and triptolide (positive control). (**A**) Binding of β-amyrin to NF-κB as determined through microscale thermophoresis (MST). The resulting binding kinetics is shown as normalized fluorescence (LED power: 40%; MST power: 10%). (**B**) Inhibition of NF-κB activity using an NF-κB reporter assay. The percentages of the NF-κB rest activity are shown after 24 h treatment with β-amyrin, β-sitosterol, β-eudesmol, daucosterol, myricetin, and apigenin at concentrations of 0.1 µM, 1 µM, and 10 µM followed by 100 ng/mL of TNF-α for 24 h (* *p* < 0.05). (**C**,**D**) Pearson correlation of NF-κB rest activity (%) vs. lowest binding energy (kcal/mol) of the six selected compounds at concentrations of (**C**) 0.1 µM and (**D**) 1 µM. The correlation using a concentration of 10 µM was statistically not significant. The mean values ± SD of three independent experiments are shown. (* *p* < 0.05, ** *p* < 0.01, *** *p* < 0.001).

**Figure 3 biomedicines-12-01484-f003:**
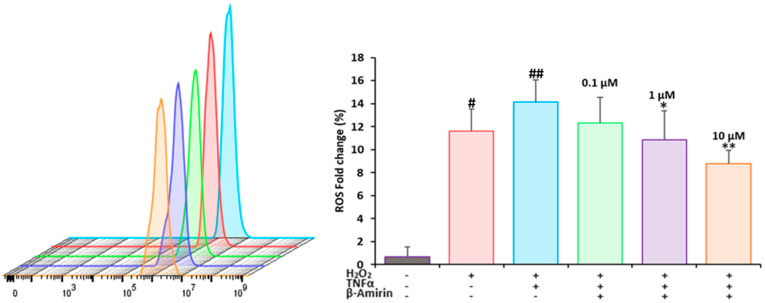
Effect of β-amyrin on the generation of reactive oxygen species (ROS) in HEK-Blue Null 1 (HBN1). The cells were treated with 100 µM of H_2_O_2_ (15 min) and 100 ng/mL of TNF-α (3 h) with and without β-amyrin at concentrations of 0.1 µM, 1 mM, and 10 µM (24 h). The statistical analysis was performed by using the paired student’s *t*-test. * *p* = 0.05 (1 µM) and ** *p* = 0.01 (10 µM) compared with TNF-α- and H_2_O_2_-treated control cells, ^#^
*p* = 0.02 (cells treated with H_2_O_2_), and ^##^
*p* = 0.005 (cells treated with TNF-α and H_2_O_2_) compared with untreated cells. β-amyrin significantly reduced ROS generation. DMSO treatment served as the solvent control. The mean values ± SD of three independent experiments are shown.

**Figure 4 biomedicines-12-01484-f004:**
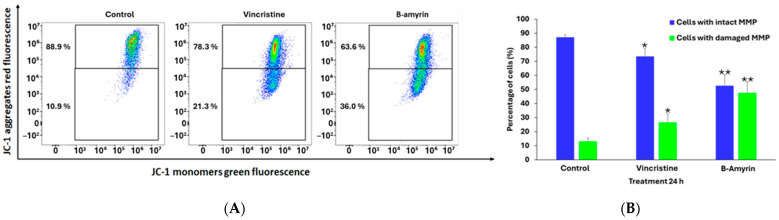
Flow cytometric determination of mitochondrial membrane potential (MMP) in HEK-Blue Null 1 Cells through JC–1 staining. Cells were left untreated (control) or treated with 10 µM of β-amyrin or vinblastine for 24 h, followed by 100 ng/mL of TNF-α for 3 h. (**A**) Representative histograms; (**B**) statistical analysis cells with disrupted MMP (dead cells) or intact MMP (healthy cells). Mean values ± SD of three independent experiments are shown. The results are significant at ** *p* < 0.01 for death cells and * *p* < 0.05 for living cells if compared to the control DMSO untreated cells (paired two-tailed *t*-test).

**Figure 5 biomedicines-12-01484-f005:**
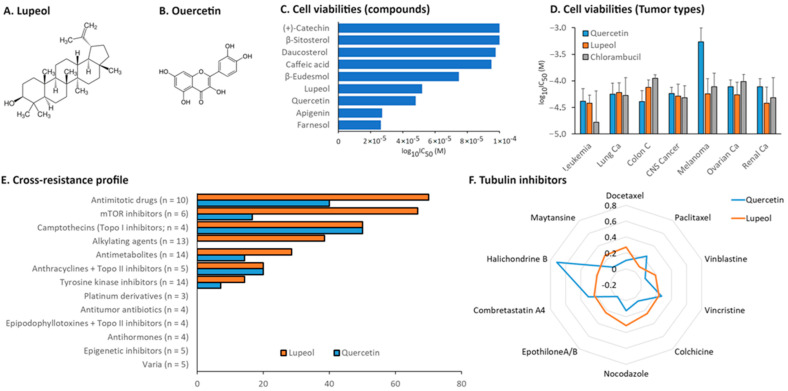
Cytotoxicity and oncobiogram analyses. Chemical structures of (**A**) lupeol and (**B**) quercetin. (**C**) Cytotoxicity of six selected phytochemicals from chamomile to the NCI tumor cell line panel plotted as a mean log_10_IC_50_ for each tumor type. (**D**) The cytotoxicity of lupeol and quercetin in cell lines of different tumor types compared with the established anticancer drug chlorambucil as the positive control. (**E**) Cross-resistance profiling of lupeol and quercetin to 91 standard drugs with known modes of action against tumor cells. (**F**) Oncobiogram for lupeol and quercetin. The correlation coefficients for lupeol and quercetin to 10 known tubulin inhibitors are plotted.

**Figure 6 biomedicines-12-01484-f006:**
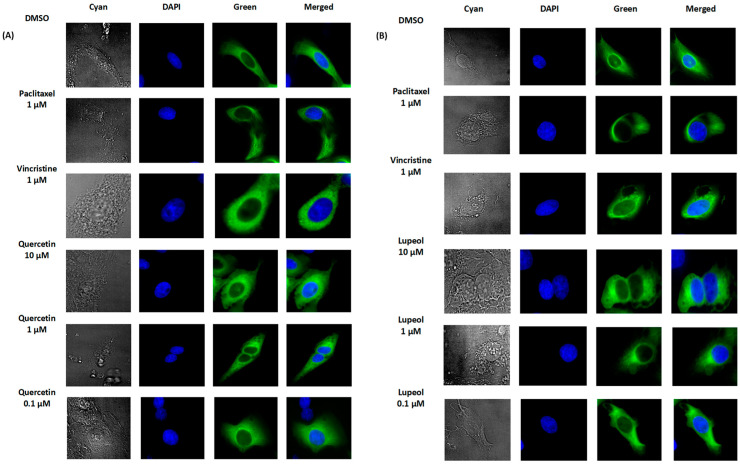
Confocal immunofluorescence microscopy of the microtubule network in U2OS cells upon treatment with (**A**) quercetin and (**B**) lupeol at concentrations of 0.1 µM, 1 µM, and 10 µM for 24 h. Vincristine (1 µM) and paclitaxel (1 µM) served as positive controls and DMSO as the negative control. The cells were imaged using a Thunder Imager Live Cell microscope with a 63×/1.40 NA objective lens (HC PL APO CS2 63×/1.40 OIL UV). The microtubules were visualized using green fluorescence for GFP (green), and the images were merged with DAPI (blue) to highlight the nucleus.

**Figure 7 biomedicines-12-01484-f007:**
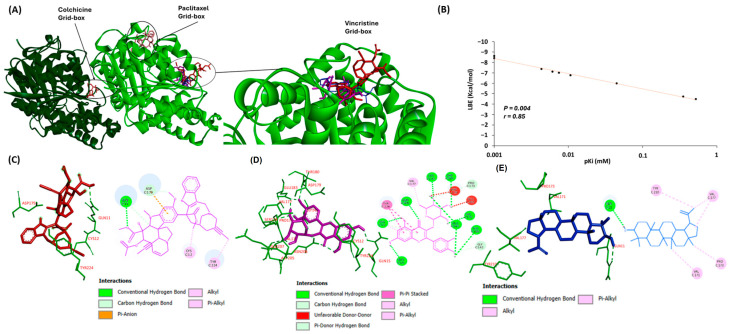
Molecular docking analysis of lupeol and quercetin to tubulin (5N5N). (**A**) Illustrates the binding sites of vincristine, paclitaxel, and colchicine to α- and β-tubulin. On the right, a zoomed-in view shows lupeol and quercetin binding to the same pocket as vincristine. (**B**) The correlation of the predicted inhibition constants (pKi, mM) vs. the lowest binding energies (LBE, kcal/mol) (*p* = 0.004, *r* = 0.85). (**C**,**D**) Presents 3D and 2D illustrations of the interaction of lupeol, quercetin, and vincristine with the amino acids of α-tubulin docked at vincristine’s binding site. (**C**) Vincristine, (**D**) quercetin, and (**E**) lupeol.

**Figure 8 biomedicines-12-01484-f008:**
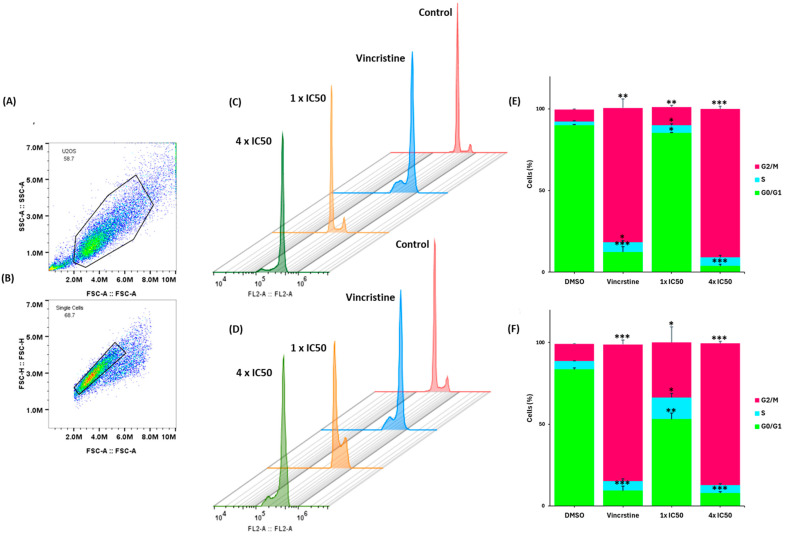
Cell cycle arrest of U2OS cells by quercetin and lupeol. (**A**) Debris was gated out (SSC-A vs. FSC-A) with the first gate. (**B**) With the second gate (FSC-H vs. FSC-A), only single cells of normal morphology were gated. Duplets were gated out. (**C**,**D**) Three-dimensional representation of DNA histograms of U2OS cells exposed to 1 × IC_50_ and 4 × IC_50_ quercetin and lupeol for 72 h. DMSO was used as the negative control, and 1 × IC_50_ vincristine was used as the positive control. (**C**) Cells treated with lupeol and (**D**) cells treated with quercetin. The histograms were obtained through flow cytometry using an excitation of 488 nm and an emission wavelength of 530 nm. (**E**,**F**) Bar diagrams showing the distinct phases of cell cycle upon treatment with quercetin and lupeol for 72 h. (**E**) Cells treated with lupeol and (**F**) cells treated with quercetin. The bar diagrams were created through the calculation of the mean values ± SD of three independent experiments. *** *p* < 0.001, ** *p* < 0.01, and * *p* < 0.05 compared to the negative control using paired two-tailed *t*-test.

**Figure 9 biomedicines-12-01484-f009:**
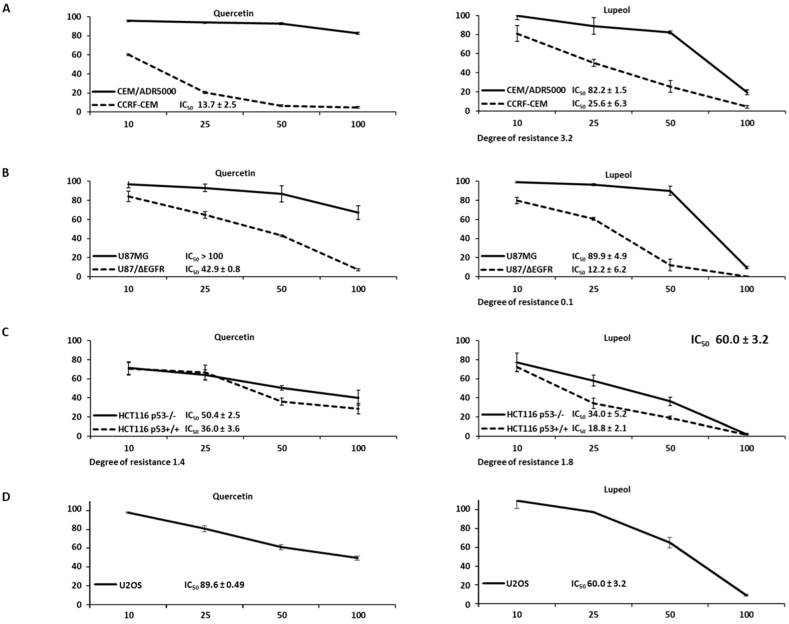
Dose–response curves of quercetin and lupeol as determined through resazurin assay. The mean values and standard deviation values are from three independent experiments. The tumor cells were subjected to treatment with each compound at concentrations of 10 µM, 25 µM, 50 µM, and 100 µM for 72 h. (**A**) Sensitive CCRF-CEM and the drug-resistant P-glycoprotein overexpressing CEM/ADR5000 leukemia cells. (**B**) U87/ΔEGFR transfected with a deletion-activated cDNA of EGFR and its wild-type U87MG glioblastoma cells. (**C**) HCT116 p53^+/+^ and knockout HCT116 p53^−/−^ colorectal cancer cells. (**D**) U2OS osteosarcoma cells.

**Figure 10 biomedicines-12-01484-f010:**
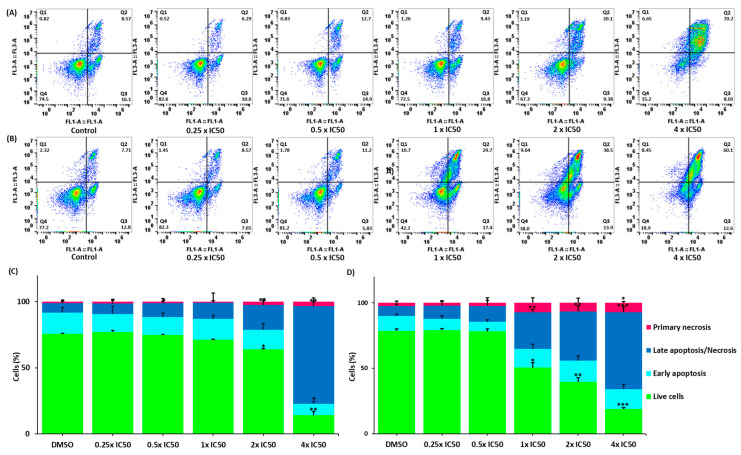
Detection of cell death in CCRF-CEM cells using flow cytometry and annexin-V/PI staining to measure apoptosis using a flow cytometer. (**A**,**B**) Cells treated with 0.25 × IC_50_, 0.5 × IC_50_, 1 × IC_50_, 2 × IC_50_, and 4 × IC_50_ of quercetin and lupeol, for 72 h. DMSO was used as negative control. (**A**) Cells treated with lupeol and (**B**) cells treated with quercetin. Q1 represents necrotic cells (−) annexin V/(+) PI; Q2 represents late apoptotic cells exhibiting annexin V (+)/PI (+); Q3 represents early apoptotic cells (+) annexin V/(−) PI; Q4 represents viable cells (−) annexin V/(−) PI. (**C**,**D**) Bar diagrams representing the percentages of cells in the different quadrants. (**C**) Effects of lupeol and (**D**) Effects of quercetin. The treatment of both compounds at increasing concentrations significantly enhanced the percentage of necrotic cells. *** *p* < 0.001, ** *p* < 0.01, and * *p* < 0.05 compared to the negative control using paired two-tailed t-test. The bar diagrams were created based on the calculation of the mean values ± SD of three independent experiments.

**Figure 11 biomedicines-12-01484-f011:**
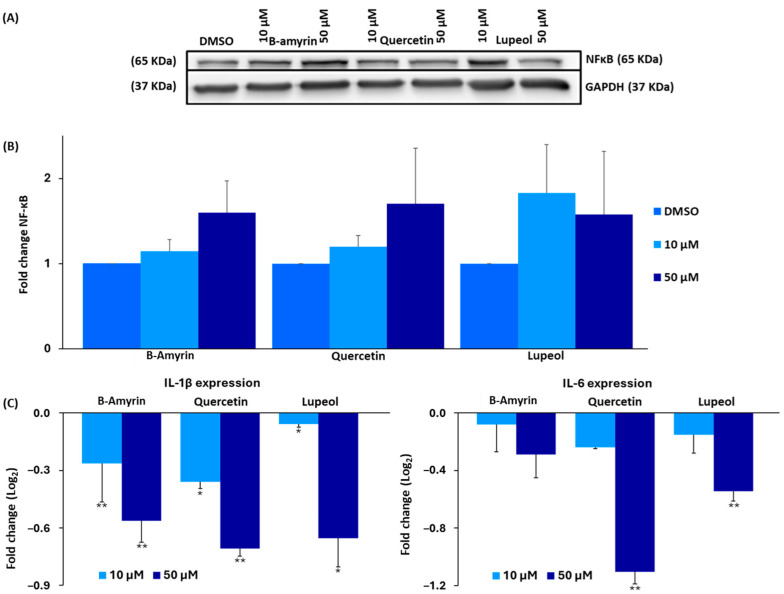
Expression of NF-κB and downstream genes. (**A**) Western blotting analysis of NF-κB in HEK-Blue Null1 cells treated with 10 µM or 50 µM of β-amyrin, quercetin, and lupeol for 24 h, followed by 24 h of TNF-α at 100 ng/mol. (**B**) The percentages of NF-κB expression in the cell. (**C**) qRT-PCR analysis of *IL-1β* and *IL-6* gene expression in HEK-Blue Null 1 cells treated with 10 µM or 50 µM of β-amyrin, quercetin, and lupeol for 24 h with TNF-α 100 ng/mL for another 24 h. The statistical analysis was performed through the paired one-tailed *t*-test (** *p* ≤ 0.01) (* *p* ≤ 0.05) from three independent trials.

**Figure 12 biomedicines-12-01484-f012:**
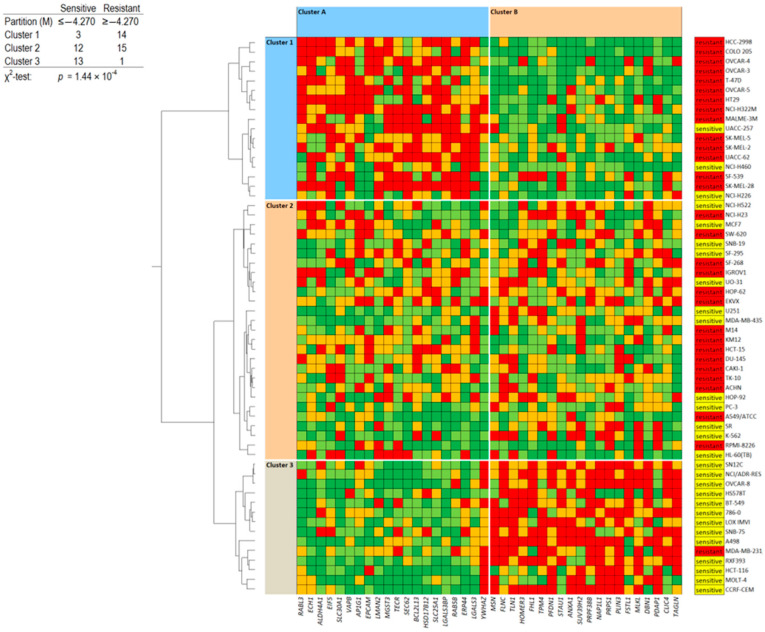
Cluster analysis in a 2D colored heat map of proteins’ expression in NCI tumor cell lines responding to lupeol. Clusters A and B represent the top 40 proteins, and clusters 1–3 show the tumor cell lines. The cell lines were clustered according to their degrees of relatedness to each other based on their protein expression included in the analysis. Color code: red, 0–25% quartile; orange, 26–50% quartile; grey, median value; light green, 50–75% quartile; and dark green, 76–100% quartile. Depending on individual log_10_IC_50_ values, the responsiveness of the cell lines to lupeol was classified as sensitive if their log_10_IC_50_ values were lower than the median value of all cell lines (marked in yellow) and as resistant if their log_10_IC_50_ values were higher than the median value (marked in red). The χ2 test shows a statistical significance (*p* = 1.44 × 10^−4^) upon comparing the two clusters of protein expression in the cell lines, where clusters 1 and 2 contained mainly resistant cell lines to both lupeol and quercetin, respectively, and cluster 3 contained mainly sensitive cell lines.

**Figure 13 biomedicines-12-01484-f013:**
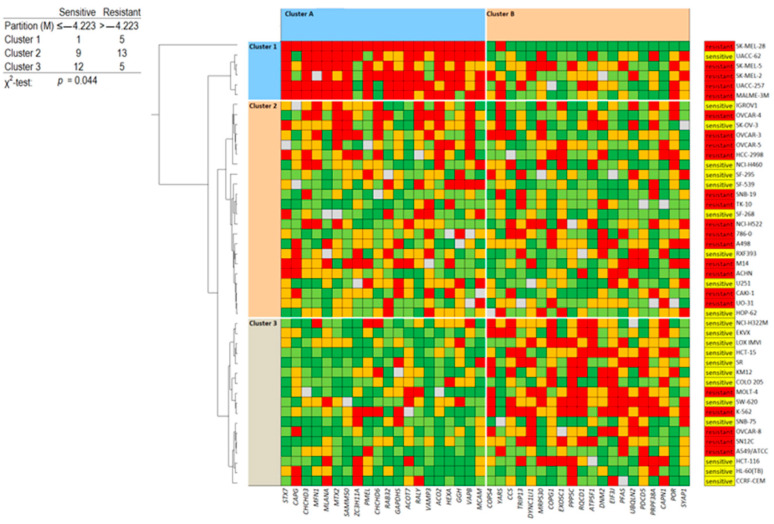
Cluster analysis in a 2D colored heat map of proteins’ expression in NCI tumor cell lines responding to quercetin. The χ2 test shows a statistical significance (*p* = 0.044). For further details, see Figure 12.

**Figure 14 biomedicines-12-01484-f014:**
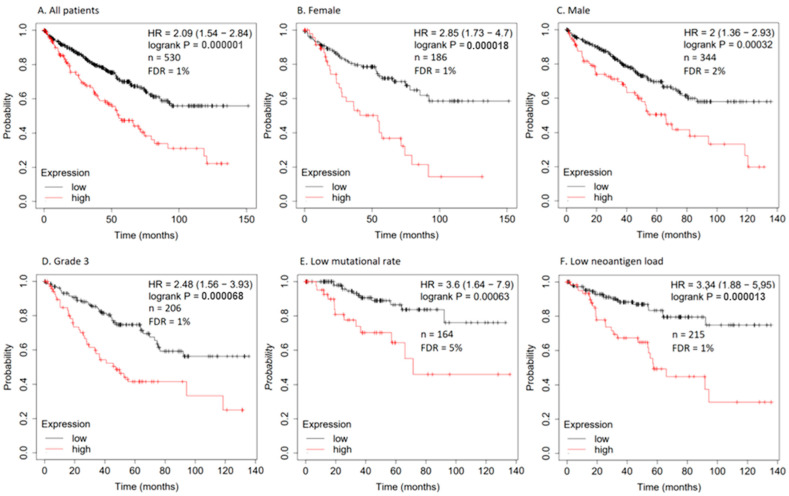
Kaplan–Meier analysis from the KM-Plotter database of overall survival time (months) for renal clear cell carcinoma cells correlating with the expression of *NFKB2* mRNA. (**A**–**F**) Different patient profiles where NF-κB expression correlated significantly with the overall survival time.

**Table 1 biomedicines-12-01484-t001:** Molecular docking of 28 chamomile compounds bound to NF-κB p65-RelA homodimer (PDB ID: 1NFI). The lowest binding energies (LBE, kcal/mol) predicted inhibition constants (µM), and the pharmacophores with the amino acid residues involved in binding of the compounds are represented.

**Compounds**	**LBE (kcal/mol)**	**pKi (μM)**	**Pharmacophores**
β-Amyrin	−8.70 ± <0.01	0.42 ± <0.01	GLU222, **ASP223**, **ILE224**, GLU225, **PHE239,** PRO275
Lupeol	−7.59 ± 0.01	2.72 ± 0.05	LYS28, GLU222, ASP223, ILE224, GLU225, PHE239, GLN241, PRO275
β-Sitosterol	−7.53 ± 0.10	3.06 ± 0.47	LYS28, GLU222, ASP223, ILE224, GLU225, PHE239, GLN241, PRO275
Luteolin-7-O-glucoside	−6.95 ± 0.03	8.05 ± 0.53	LYs28, ARG30, GLU222, ASP223, ILE224, GLU225, GLY237, **SER238**, **PHE239**, GLN241, PRO275, SER276
Daucosterol	−6.90 ± 0.11	8.79 ± 1.70	HIS181, GLN220, LYS221, GLU222, ALA242, **VAL244**, **ARG246**, GLN247
β-Eudesmol	−6.70 ± 0.01	12.23 ± 0.15	LYS221, GLU222, ILE224, GLU225, VAL226, ARG236, GLY237, SER238, **PHE239**, GLN241, PRO275
(−)-Epicatechin	−6.41 ± 0.04	20.01 ± 1.40	GLU222, ASP223, **ILE224**, GLU225, **PHE239**, PRO275
Myricetin	−6.32 ± 0.17	23.87 ± 7.32	THR71, ARG73, **GLU101**, **ASN139**, **PRO140**, **GLN142**, GLN162, VAL163, **THR164**, PRO177
(+)-Catechin	−6.28 ± 0.02	25.16 ± 0.95	**GLN29**, VAL219, **GLN220**, **LYS221**, VAL224, ARG246, GLN247
Quercetin hydrate	−6.15 ± 0.01	31.17 ± 0.19	**VAL219**, GLN220, LYS221, VAL244, ARG246, **GLN247**
Luteolin	−6.14 ± 0.02	32.14 ± 0.15	HIS181, GLN220, **LYS221**, **GLU222**, HIS245, **ARG246**
Kaempferol	−6.10 ± 0.01	34.24 ± 0.70	VAL219, GLN220, **LYS221**, **VAL244**, ARG246, **GLN247**
Bisabelol oxide B	−6.03 ± 0.01	38.18 ± 0.21	GLN220, LYS221, **VAL244**, HIS245, **ARG246**, GLN247
Chlorogenic acid	−6.01 ± 0.12	39.73 ± 7.82	**GLN29**, PHE184, **LYS218**, VAL219, GLN220, **LYS221**, VAL244, **GLN247**
Apigenin	−5.91 ± 0.00	46.78 ± 0.12	**GLN29**, **GLN220**, LYS221, GLU222, **VAL244**, HIS245, **ARG246**, **GLN247**
A-Bisabolol	−5.85 ± 0.03	51.77 ± 2.68	GLU222, ASP223, ILE224, GLU225, VAL226, ARG236, GLY237, SER238, **PHE239**, GLN241, PRO275
Guaiazulene	−5.75 ± <0.01	61.15 ± 0.10	GLU222, ILE224, GLU225, VAL226, ARG236, GLY237, PHE239, SER240, GLN241, PRO275
Quercitrin	−5.66 ± 0.06	70.71 ± 7.19	**VAL219**, **GLN220**, **LYS221**, **GLU222**, **VAL244**, ARG246, GLN247
Caffeic acid	−5.64 ± 0.05	73.56 ± 5.42	**VAL219**, GLN220, **LYS221**, **GLU222**, VAL244, **ARG246**, GLN247
Bisabolol oxide A	−5.37 ± <0.01	116.55 ± 0.01	VAL219, GLN220, LYS221, ALA242, **VAL244**, HIS245, **ARG246**, GLN247
Chamazulene	−5.32 ± <0.01	126.69 ± 0.26	GLU222, ASP223, ILE224, ARG236, PHE239, SER240, GLN241, PRO275
Bisabolone oxide A	−5.12 ± 0.01	175.23 ± 0.16	GLU222, ASP223, ILE224, GLU225, VAL226, ARG236, GLY237, SER238, **PHE239**, PRO275
Syringic acid	−4.96 ± 0.01	232.16 ± 2.85	**ILE23**, ILE24, GLU25, **GLN26**, GLU49, **ARG50**, **LYS221**, GLU222
Farnesol	−4.68 ± 0.02	374.02 ± 10.22	**GLN29**, **GL220**, LYS221, GLU222, ALA242, VAL244, ARG246, GLN247
Gentisic acid	−4.48 ± 0.01	518.79 ± 9.09	ILE24, GLU25, **GLN26**, **ARG50**, **LYS221**, GLU222
(+)-Terpinen-4-ol	−4.14 ± 0.01	925.85 ± 10.48	VAL219, GLN220, **LYS221**, **GLU222**, VAL244, GLN247
P-Cymene	−4.14 ± <0.01	929.21 ± 0.34	LYS221, ILE224, GLU225, ARG236, GLY237, PHE239, GLN241
Citronellol	−4.10 ± <0.01	990.93 ± 1.02	LYS221, GLU222, ILE224, GLU225, VAL226, ARG236, GLY237, SER238, **PHE239**, GLN241

**Table 2 biomedicines-12-01484-t002:** Molecular docking of lupeol and quercetin bound to α- and β-tubulin (PDB ID: 5N5N). The lowest binding energies (LBE, kcal/mol), the predicted inhibition constants (µM), and the pharmacophores are represented.

Grid Box	Compounds	LBE (kcal/mol)	pKi (mM)	Pharmacophores
Colchicine grid box	Colchicine	−7.01 ± 0.14	0.007 ± <0.01	α: LEU248, **LYS254**, LYS352 β: GLN11, ASN101, GLY142, GLY143, GLY144, THR145, GLU183, ASN206, TYR224
	Lupeol	−4.48 ± 0.08	0.520 ± 0.07	α: LEU248, LYS254, LYS352 β: GLN11, GLU71, ASP98, ALA99, ALA100, ASN101, GLY142, GLY143, GLY144, THR145, THR179, GLU183
	Quercetin	−4.72 ± 0.15	0.352 ± 0.08	α: GLN247, LEU248, LYS254, LYS352 β: GLN11, ALA12, ASP69, GLU71, ALA99, ALA100, ASN101, GLY144, THR145, THR179, ALA180
Paclitaxel grid box	Paclitaxel	−7.37 ± 0.25	0.004 ± <0.01	α: LEU217, HIS229, LEU230, ALA233, SER236, PHE272, ALA273, PRO274, LEU275, THR276, SER277, ARG278, ARG320, PRO360, **ARG369**, LEU371
	Lupeol	−7.12 ± 0.01	0.006 ± <0.01	α: THR276, GLN281, ARG284, LEU286, LEU371, LYS372
	Quercetin	−5.99 ± 0.29	0.045 ± 0.02	α: LEU217, PRO274, LEU275, THR276, SER277, ARG278, GLN281, LEU286, LEU371, **LYS372**
Vincristine grid box	Vincristine	−8.42 ± 0.15	0.001 ± <0.01	α: **GLN11**, CYS12, GLN15, ASN101, **SER140**, GLY142, GLY143, VAL172, PRO173, SER174, ASP179, THR180, ASN206, TYR224, ASN228
	Lupeol	−8.62 ± 0.04	0.001 ± <0.01	α: GLY10, GLY11, SER140, GLY142, GLY143, VAL171, VAL172, PRO173, SER174, VAL177, ASP179, GLU183, ASN206, TYR210, TYR224
	Quercetin	−6.77 ± 0.06	0.011 ± <0.01	α: CYS12, GLN15, GLY142, VAL172, PRO173, SER174, VAL177, ASP179, GLU183, **ASN206**, GLU207

**Table 3 biomedicines-12-01484-t003:** Correlation between the log_10_IC_50_ values for quercetin and lupeol and classical drug resistance mechanisms in the NCI tumor cell line panel.

		Lupeol	Quercetin	Control Drug
		(log_10_IC_50_, M)	(log_10_IC_50_, M)	(log_10_IC_50_, M)
*ABCB1* Expression				Epirubicin
7q21 (Chromosomal	*r*-value	0.160	0.128	* 0.447
Locus of *ABCB1* Gene)	*p*-value	0.121	0.203	* 3.55 × 10^−4^
*ABCB1* Expression	*r*-value	−0.132	−0.631	* 0.533
(Microarray)	*p*-value	0.159	0.339	* 6.82 × 10^−6^
*ABCB1* Expression	*r*-value	0.027	−0.033	* 0.410
(RT-PCR)	*p*-value	0.425	0.413	* 1.54 × 10^−3^
*ABCB5* Expression				Maytansine
*ABCB5* Expression	*r*-value	−0.050	0.207	* 0.454
(Microarray)	*p*-value	0.353	0.086	* 6.67 × 10^−4^
*ABCB5* Expression	*r*-value	−0.027	*0.306	* 0.402
(RT-PCR)	*p*-value	0.420	*0.021	* 0.0026
*ABCC1* Expression				Vinblastine
DNA Gene	*r*-value	0.059	−0.067	* 0.429
Copy Number	*p*-value	0.329	0.331	* 0.001
*ABCC1* Expression	*r*-value	0.040	−0.213	* 0.398
(Microarray)	*p*-value	0.383	0.082	* 0.003
*ABCC1* Expression	*r*-value	−0.023	−0.118	0.299
(RT-PCR)	*p*-value	0.436	0.207	* 0.036
*ABCG2* Expression				Pancratistatin
*ABCG2* Expression	*r*-value	0.105	−0.038	* 0.329
(Microarray)	*p*-value	0.219	0.402	* 0.006
ABCG2 Expression	*r*-value	0.010	−0.040	* 0.346
(Western Blotting)	*p*-value	0.229	0.3982	* 0.004
*EGFR* Expression				Erlotinib
*EGFR* Gene	*r*-value	0.049	−0.037	−0.245
Copy Number	*p*-value	0.357	0.404	* 0.029
*EGFR* Expression	*r*-value	−0.034	−0.068	* −0.458
(Microarray)	*p*-value	0.399	0.328	* 1.15 × 10^−4^
*EGFR* Expression	*r*-value	−0.101	0.049	* −0.379
(PCR Slot Blot)	*p*-value	0.227	0.358	* 0.002
EGFR Expression	*r*-value	−0.190	0.015	* −0.376
(Protein Array)	*p*-value	0.077	0.461	* 0.001
*N-/K-/H-RAS* Mutations				Melphalan
*TP53* Mutation	*r*-value	−0.034	0.134	* 0.367
(cDNA Sequencing)	*p*-value	0.399	0.190	* 0002
*TP53* Mutation				5-Fluorouracil
*TP53* Mutation	*r*-value	−0.036	0.395	* −0.502
(cDNA Sequencing)	*p*-value	−0.015	0.277	* 3.50 × 10^−5^
TP53 Function	*r*-value	0.050	−0.079	* −0.436
(Yeast Functional Assay)	*p*-value	0.360	0.311	* 5.49 × 10^−4^
*WT1* Expression				Ifosfamide
WT1 Expression	*r*-value	−0.046	−0.103	* −0.316
(Microarray)	*p*-value	0.365	0.250	* 0.007
*GSTP1* Expression				Etoposide
*GSTP1* Expression	*r*-value	−0.012	−0.012	0.399
(Microarray)	*p*-value	0.468	0.468	* 9.58 × 10^−4^
*GST* Expression	*r*-value	*0.302	0.028	0.509
(Northern Blot)	*p*-value	*0.010	0.427	* 2.24 × 10^−5^
*HSP90* Expression				Geldanamycin
*HSP90* Expression	*r*-value	−0.055	−0.045	* −0.392
(Microarray)	*p*-value	0.342	0.384	* 0.001
Proliferation				5-Fluorouracil
(Cell Doubling)	*r*-value	−0.185	0.076	* 0.627
	*p*-value	0.084	0.313	* 7.14 × 10^−6^

(* indicate *p* < 0.05 and *r* > 0.30 or *r* < −0.30).

**Table 4 biomedicines-12-01484-t004:** Functional categories of proteins identified through the proteomic analyses for lupeol and quercetin, as shown in Figure 12 and Figure 13.

Functional Categories	Protein Symbols (Lupeol Analysis)	Protein Symbols (Quercetin Analysis)
General metabolism	ALDH4A1, ECH1, HSD17B12, TECR	ACO2, ACOT7, GAPDHS, GGH
Protein and vesicle trafficking	AP1G1, LMAN2, PLIN3, RAB5B, SEC62, VAPB	COPG1, DYNC1LI1, STX7, VAPB
DNA/RNA metabolism	EIF5, FHL1, NAP1L1, STAU1, SUV39H2	DYNC1LI1, EIF3J, EXOSC1, PDCD5, PRPF38A, RALY, TRIP13, ZC3H11A
Cell Adhesion	EPCAM, LGALS3, TLN1	MCAM
Chaperones and protein degradation	ERP44, PFDN1	CCS, UBQLN2
Cytoskeleton	FLNC, MSN, STAU1, TPM4	CAPG, DNM2, CAPN1
Immune function	ANXA1, FSTL1, LGALS3, LGALS3BP, MGST3, RAB5B, TLN1	YARS
Signal transduction	HOMER2, RABL3, YWHAZ	CAPN1, PPP5C, RQCD1
Cell proliferation and differentiation	DBN1, PDAP1	PDCD5, PPP5C, RQCD1
Cell death	BCL2L13, LGALS3, MLKL	PDCD5, RAB32
Tumor suppressor	TAGLN	
Ion channels and drug transporters	CLIC4, SLC25A1, SLC30A1	
Mitochondrial function		ACO2, ATP5F1, CHCHD3, CHCHD6, MFN1, MRPS30, MTX2, RAB32, SAMM50
Melanosome biogenesis and function		MLANA, PMEL, RAB32
Drug metabolism		POR
Others		CHCHD6, HEXA, VAMP3

(Multiple entries are possible).

## Data Availability

Data are available upon reasonable request.

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
