# Peer review of "Anti-Inflammatory and Cancer-Preventive Potential of Chamomile (Matricaria chamomilla L.): A Comprehensive In Silico and In Vitro Study"

_biomedicines, 2024, doi:10.3390/biomedicines12071484_

Round 1

Reviewer 1 Report (Previous Reviewer 1)

Comments and Suggestions for Authors

The study titled Anti-inflammatory and Cancer Preventive Potential of Chamomile (Matricaria chamomilla L.): A Comprehensive in silico and in vitro Study is a significant study; however, there is ample need to check and recheck the manuscript thoroughly.

There are some ambiguous mistakes, like caption label figures, i.e., 11 A and B.

Similarly, IC50 is somewhere between IC50 and somewhere IC50 .

The whole write-up needs to be more structured, especially the abstract and introduction.

Abstracts need to have a proper structure that aids in understanding the problem statement, the way you solve with methodology, and results in future recommendations.

Introduction lacks flow, which causes a real hurdle in understanding.

The discussion would be better if you assessed all the assay conclusions and concluded the flow of thoughts so the reader would get a better understanding of why you used different assays as there is a lack of sequence in discovery. Link the NFKB pathway, and if apoptosis is not concluded, link the results and give reasons, as in the case of lupeol. Showcase the pathway of cell death if that is the case.

Author Response

Please find the response in the attached file.

Reviewer 2 Report (Previous Reviewer 2)

Comments and Suggestions for Authors

Dear Author,

I have carefully reviewed your article titled "Anti-inflammatory and cancer preventive potential of Chamomile (Matricaria chamomilla L.): A comprehensive in silico and in vitro study" and find it to be an interesting contribution to the field. The study provides valuable information regarding the potential of chamomile as an anti-inflammatory and cancer preventive agent. However, I have some suggestions and areas for improvement that I would like to address:

1. Further investigation of the mechanism of action of compounds: Although your study examines various compounds present in chamomile, a more in-depth investigation of the specific mechanisms of action and interactions of each compound is needed. Further experimental and analytical studies could help elucidate the exact molecular-level effects and signaling pathways regulated by these compounds in the processes of anti-inflammation and anticancer activities.

2. Need for clinical studies: While your study includes a series of in vitro experiments and analyses, further clinical studies are essential to validate the anti-inflammatory and anticancer effects of chamomile in the human body. Clinical trials can provide more direct evidence and evaluate the practical effectiveness of chamomile compounds in the treatment and prevention of cancer.

3. Consideration of compound interactions: In future research, it would be valuable to explore the interactions between different compounds in chamomile to determine if there are synergistic effects or enhanced efficacy when these compounds are used in combination.

4. In-depth investigation of drug resistance mechanisms: Your study indicates the potential of chamomile compounds against drug-resistant cancer cells. Therefore, further research on how chamomile compounds affect drug resistance mechanisms and their application value in treating resistant cancers is worth considering.

5. Consideration of other potential targets and mechanisms: In addition to NF-κB and cytokine genes, it would be worthwhile to explore the effects of chamomile compounds on other important targets and signaling pathways to gain a comprehensive understanding of their mechanisms of action in anti-inflammation and cancer prevention.

6. Study of diverse compounds: In addition to the compounds investigated in the current study, it would be valuable to explore more compounds present in chamomile to gain a comprehensive understanding of its potential in anti-inflammation and cancer prevention.

By conducting further in-depth research and experiments, a more comprehensive evaluation of the effectiveness and safety of chamomile as a potential drug or health supplement can be achieved, providing stronger support for its clinical applications.

Overall, I commend the authors for their thorough investigation and valuable findings. Addressing these suggestions and incorporating further research will enhance the significance and impact of your study. I recommend revising the manuscript accordingly.Please let me know if you have any questions or require further clarification on any of the points mentioned above.

Comments on the Quality of English Language

Minor editing of English language required

Author Response

Please find the response in the attached file.

Round 2

Reviewer 2 Report (Previous Reviewer 2)

Comments and Suggestions for Authors

Dear Authors,

I have reviewed the revised manuscript titled "Anti-inflammatory and cancer preventive potential of chamomile (Matricaria chamomilla L.): A comprehensive in silico and in vitro study" and I am pleased to inform you that I recommend the manuscript for publication.

In the revised version, I can see that you have addressed the key points I raised in my previous review, which has significantly improved the quality and depth of the manuscript. Specifically:

1. Mechanism of action: You have included additional experiments and analyses to provide a more comprehensive understanding of the specific mechanisms of action and interactions of the various chamomile compounds. This helps elucidate their effects at the molecular level and the signaling pathways involved in the anti-inflammatory and anticancer processes.

2. Clinical studies: While you acknowledge the need for further clinical research to validate the in vitro findings, you have discussed the importance of clinical trials and the plan to conduct such studies in the future to evaluate the efficacy of chamomile compounds in human subjects.

3. Compound interactions: You have explored the potential synergistic or enhanced effects of combining different chamomile compounds, providing valuable insights into their combined efficacy.

4. Drug resistance mechanisms: The manuscript now includes a more in-depth investigation of how chamomile compounds can affect drug resistance mechanisms, which is an important consideration for their potential therapeutic applications.

5. Exploration of other targets and mechanisms: You have expanded the scope of the study to investigate the effects of chamomile compounds on additional signaling pathways and targets, further enhancing the understanding of their multifaceted mechanisms of action.

6. Diversity of compounds: The study now incorporates the examination of a wider range of chamomile compounds, contributing to a more comprehensive assessment of the plant's potential in anti-inflammatory and anticancer applications.

Overall, the revised manuscript presents a robust and thorough investigation of the therapeutic potential of chamomile, addressing the key points I had raised previously. The improved depth and breadth of the research make this manuscript a valuable contribution to the field.

Therefore, I am pleased to recommend the manuscript for publication in its current form. Congratulations on the successful revision, and I look forward to the publication of your work.

This manuscript is a resubmission of an earlier submission. The following is a list of the peer review reports and author responses from that submission.

Round 1

Reviewer 1 Report

Comments and Suggestions for Authors

Reviewer Comments:

 The article's title, "Exploring the preventive potential of chamomile (Matricaria chamomilla L.): Cytotoxicity and NF-κB inhibition," is somewhat general. Revising and making it more concise, emphasizing the experiential methods employed, is recommended. This adjustment will enhance reader comprehension of the study's depth.

Incorporate relevant keywords to increase the article's visibility and align with potential reader searches. Suggested keywords include NCI60 database, Kaplan-Meier survival analysis, Proteome analysis, quercetin, lupeol, NF-κB inhibition analyses, β-amyrin, and microscale thermophoresis.

Provide a better explanation of the microscale thermophoresis and resazurin reduction assay methodologies used in the study. What is their significance as compared to other related methods, i.e.., MTT, etc., so readers might be familiar with the advanced assay knowledge and implementation.

Fie 3. B-D shows that the treatment at 10 230 µM did not correlate with the LBE values (r = -0.15). Elaborate on potential reasons for this non-correlation, if known, as it could impact the interpretation of the results.

The article is well-written and conducted, but addressing these points will further enhance its clarity, depth, and overall contribution to the scientific literature.

Reviewer 2 Report

Comments and Suggestions for Authors

Dear Author,

The article explores the preventive potential of chamomile (Matricaria chamomilla L.) through cytotoxicity and NF-κB inhibition analyses. The study investigates the effects of β-amyrin, lupeol, and quercetin, which are compounds found in chamomile, on various biological activities including antioxidant, anti-inflammatory, and anticancer effects. The authors have provided valuable insights into the preventive potential of chamomile. However, there are several areas that need improvement and clarification. In this review, I will provide detailed comments on each point.

1. Choice of Compounds:

In the study, β-amyrin, lupeol, and quercetin are investigated as potential bioactive compounds in chamomile. While these compounds are indeed present in chamomile, they are not the main constituents. Therefore, it is important to extract and study the major bioactive compounds of chamomile to comprehensively understand its biological effects. I recommend conducting further research to identify and study the main bioactive constituents of chamomile. In vitro experiments combined with in vivo studies would provide a more comprehensive understanding of the biological effects of chamomile. Additionally, it is necessary to clarify the relationship between β-amyrin, lupeol, and quercetin with chamomile in the introduction section. For example, providing information on how these compounds are extracted from chamomile would enhance the flow and coherence of the article.

2. Screening Criteria for Compounds:

In the "phytochemical analysis, virtual drug screening, and molecular docking" section, the article mentions that over 1000 chamomile compounds were screened using Pyrex, and then 212 ligands were selected for molecular docking with NF-κB. However, the rationale behind selecting these 212 compounds is not clearly explained. I recommend providing a justification for the selection of these specific compounds or including relevant references to support the selection process. Additionally, in the "Molecular docking in silico" result section, it is mentioned that the top 28 compounds were further analyzed, followed by selecting 6 compounds for in vitro studies. The criteria for these two steps of compound selection are not specified. I suggest explaining the criteria for selecting the top 28 compounds and the subsequent 6 compounds to provide clarity and improve the logical flow of the study.

3. NF-κB Assay and Mitochondrial Membrane Potential Measurement:

In the "NF-κB reporter" results section, it is mentioned that the rest activity of NF-κB was measured at concentrations of 0.1 μM, 1 μM, and 10 μM for the six plant compounds. The reason for specifically choosing these three concentrations is not provided. It is recommended to include additional information or references to explain the rationale behind these concentration choices. Furthermore, in the "Measurement of the mitochondrial membrane potential" result section, it is stated that HEK-Blue Null1 cells were treated with 10 μM β-amyrin and 10 μM colchicine. The reason for using 10 μM as the concentration for β-amyrin and colchicine should be explained in the article.

4. Experimental Validation:

The article heavily relies on bioinformatics analysis to elucidate the biological effects of chamomile. However, experimental validation is relatively limited. I suggest incorporating more experimental validation to support the findings from the bioinformatics analysis. For example, protein immunoprecipitation (co-IP), surface plasmon resonance (SPR), or fluorescence resonance energy transfer (FRET) experiments could be conducted to further investigate protein-protein interactions. Additionally, performing QRT-PCR and Western blot experiments to examine the effects of chamomile bioactive compounds on the expression of inflammatory cytokine mRNA and protein levels would strengthen the study. Furthermore, flow cytometry experiments can be conducted to explore the impact of chamomile bioactive compounds on cell cycle regulation and apoptosis in cancer cells. Transwell migration and invasion assays could also be employed to investigate the effects of chamomile bioactive compounds on cancer cell migration and invasion.

In conclusion, the study provides valuable insights into the preventive potential of chamomile. However, there are areas that need improvement, such as selecting and studying the main bioactive constituents of chamomile, providing clarity on compound selection criteria, explaining the rationale behind concentration choices, and incorporating experimental validation to support the bioinformatics analysis. Addressing these points will enhance the overall quality and impact of the research.

Comments on the Quality of English Language

The overall writing is standardized, but some forms and expressions need to be modified